# Photocatalytic CO₂ reduction to syngas using metallosalen covalent organic frameworks

Wei Zhou[1], Xiao Wang[1], Wenling Zhao[1], Naijia Lu[1], Die Cong[1], Zhen Li [1], Peigeng Han[1], Guoqing Ren [1], Lei Sun [1], Chengcheng Liu [1] ✉ & Wei-Qiao Deng [1] ✉

Metallosalen-covalent organic frameworks have recently gained attention in photocatalysis. However, their use in CO₂ photoreduction is yet to be reported. Moreover, facile preparation of metallosalen-covalent organic frameworks with good crystallinity remains considerably challenging. Herein, we report a series of metallosalen-covalent organic frameworks produced via a one-step synthesis strategy that does not require vacuum evacuation. Metallosalen-covalent organic frameworks possessing controllable coordination environments of mononuclear and binuclear metal sites are obtained and act as photocatalysts for tunable syngas production from CO₂. Metallosalen-covalent organic frameworks obtained via one-step synthesis exhibit higher crystallinity and catalytic activities than those obtained from two-step synthesis. The optimal framework material containing cobalt and triazine achieves a syngas production rate of 19.7 mmol g⁻¹ h⁻¹ (11:8 H₂/CO), outperforming previously reported porous crystalline materials. This study provides a facile strategy for producing metallosalen-covalent organic frameworks of high quality and can accelerate their exploration in various applications.

Covalent organic frameworks (COFs) are a burgeoning class of porous organic polymers that have garnered considerable attention in various fields, such as catalysis, gas adsorption and separation, energy storage and sensing[1,2]. Metallo-covalent organic frameworks (M-COFs) represent a fusion of organometallic and polymer chemistry that might be instrumental in creating excellent heterogeneous single-metal-site catalysts (SMS catalysts)[3]. In M-COFs, active metal sites are periodically integrated into extended frameworks that can effectively protect metal sites from aggregation[4–9]. Furthermore, well-defined coordination environments of metals can be conveniently achieved in M-COFs[10,11]. Determining the precise position and microenvironments of metal species in M-COFs is useful for understanding the relationships between structures and properties in catalysis. Subsequently, the catalytic performance of M-COFs can be optimised by regulating the metal species and their coordination environments[12–15]. Salen, porphyrins, and 2, 2'-bipyridine (bipy) are three chelating ligands commonly used to coordinate metal ions that form metal complexes.

Among them, the salen moiety is one of the most essential ligands in coordination chemistry because of its ability to form various metal complexes (M(salen)) that can be utilised as the most powerful homogeneous catalysts for various organic transformations[15]. The metallosalen-covalent organic frameworks (M(salen)-COFs), which embed M(salen), retain both the unique catalytic activities of M(salen) and the extended porous frameworks of COFs, endowing them with promising capabilities of heterogeneous SMS catalysts[3–6,16–18]. However, using M(salen)-COFs in photocatalytic CO₂ reduction has not been reported.

Furthermore, the synthesis of M(salen)-COFs usually involves at least a two-step process that requires post-synthetic metalation[5,6,19]. The metalation procedure inevitably results in increased time consumption, lower yields and decreased COF crystallinity due to the framework's collapse[20]. Moreover, producing highly crystalline M(salen)-COF requires high-purity multidentate organic ligands, which increases the difficulty of synthesis. In most cases, the ampoule

[1]Institute of Frontier and Interdisciplinary Science, Shandong University, 266237 Qingdao, Shandong, China. ✉e-mail: chengcheng.liu@sdu.edu.cn; dengwq@sdu.edu.cn

containing ligands, solvents and catalysts must be flash-frozen in a liquid nitrogen bath, evacuated, and flame-sealed[21]. In general, only tens of milligrams of highly crystalline M(salen)-COFs can be obtained. Harsh reaction conditions and low yields severely limit large-scale production and exploration for various applications. Thus, developing a facile synthetic method for M(salen)-COFs with high yield is vital and an attractive strategy is a multicomponent one-step reaction with more reversible chemistry[22–24].

Herein, we report the preparation of M(salen)-COFs via a one-step synthesis in autoclaves without vacuum deaeration, thus yielding gram-scale M(salen)-COFs. We fabricated mononuclear metal site M(salen)-COFs with metal coordination environments of M-$N_2O_2$ (e.g. Zn-TAPB-COF-1) and binuclear metal site M(salen)-COFs with metal coordination environments of M-$N_2O_2$-M-$N_2O_2$ (e.g. ZnZn-TAPB-COF-1). We also allowed Co instead of Zn to be embedded in the frameworks. The precise position and coordination environments for the metal sites were determined via X-ray absorption fine structure (XAFS) of synchrotron radiation analysis. Finally, the synthesised M(salen)-COFs were applied as photocatalysts in $CO_2$ reduction. Changing the metal species, coordination environments, and ligands in M(salen)-COFs could easily achieve tunable syngas proportions. In particular, the M(salen)-COFs obtained via the one-step strategy exhibited considerably stronger crystallinity and higher photocatalytic activities than those obtained via two-step synthesis.

## Results

### Synthesis and structural characterisation of M(salen)-COFs

We conducted two synthesis routes for M(salen)-COFs from the simple alicylaldehyde ligand, i.e. one-step and two-step syntheses (Fig. 1). The one-step synthesis can construct M(salen)-COFs via four-component reactions and exhibits remarkable advantages over the two-step synthesis in terms of simplicity. For the one-step synthesis of M(salen)-COFs-1, 4-hydroxyisophthalaldehyde (or 2-hydroxybenzene-1, 3, 5-tricarbaldehyde), ethylenediamine, metal acetate (Fig. 1a) and tridentate amine ligands (Fig. 1b) were first added into the autoclave in a random order. Further, ethanol, 1,2-dichlorobenzene (o-DCB), N, N-dimethylformamide (DMF) and an acetic acid (3 M HOAc) mixture were added as reaction solvents and catalysts. The autoclave was further heated under solvothermal conditions (120 °C, 72 h). For the two-

step synthesis of M(salen)-COFs-2, M(salen)-CHO or MM(salen)-CHO (M = Zn and Co) were first synthesised (Fig. 1c and Figs. S1–S4). Subsequently, M(salen)-CHO and the corresponding tridentate amine ligands (Fig. 1c), including 1, 3, 5-tris(4-aminophenyl)benzene (TAPB) or 2, 4, 6-tris(4-aminophenyl)−1, 3, 5-triazine (TAPT), were mixed and heated in the solvent system HOAc/o-DCB/n-BuOH at 120 °C for 3 days. M(salen)-COFs were obtained after washing with ethanol and tetrahydrofuran (THF). Further, the wet M(salen)-COFs were soaked in methanol and liquid $CO_2$ (7.5 MPa, 25 °C) for 24 h. Finally, the M(salen)-COFs were dried with supercritical $CO_2$ (scCO$_2$, 8 MPa, 35 °C) for 24 h. The M(salen)-COFs yields obtained via the one-step route were higher than those obtained via the two-step route (Table S1). Zn-TAPB-COFs and ZnZn-TAPB-COFs were investigated in detail as representative structures of mononuclear metal site M(salen)-COFs (Fig. 2a) and binuclear metal site M(salen)-COFs (Fig. 2b). Apart from Zn-COFs, Co-COFs were also synthesised via one- and two-step synthesis routes (Fig. 2c).

The crystalline structures of these M(salen)-COFs were characterised using powder X-ray diffraction (PXRD) technology. All M(salen)-COFs obtained from the one-step synthesis showed sharp diffraction peaks (Fig. 2g–i). Simulated eclipsed AA-stacking, serrated AA-stacking, inclined AA-stacking and AB-stacking models for Zn-TAPB-COF-1, ZnZn-TAPB-COF-1 and Co-TAPT-COF-1 were constructed using Materials Studio software (Fig. 2g–i, Figs. S5–S6 and Supplementary Data 1). The experimental PXRD patterns of the three M(salen)-COFs were all more consistent with their eclipsed AA-stacking and serrated AA-stacking models[25]. Given that only 1D fringes were observed and the honeycomb structures were absent in their TEM images, AA serrated models without 6-fold or 3-fold symmetry were more reasonable modes for the synthesised M(salen)-COFs. Furthermore, the unit-cell parameters of M(salen)-COFs-1 were optimised by a universal force field (Zn-TAPB-COF-1: $a = 51.444(1)$ Å, $b = 51.235(8)$ Å, $c = 7.373(2)$ Å, $\alpha = 74.6458°$, $\beta = 98.9531°$, $\gamma = 121.1356°$; ZnZn-TAPB-COF-1: $a = 51.291(3)$ Å, $b = 51.194(2)$ Å $c = 7.137(7)$ Å, $\alpha = 88.8966°$, $\beta = 92.1975°$, $\gamma = 119.7291°$; Co-TAPT-COF-1: $a = 50.573(2)$ Å, $b = 50.127(7)$ Å, $c = 7.087(5)$ Å, $\alpha = 74.4833°$, $\beta = 97.2283°$, $\gamma = 120.9587°$)[20]. In addition, fractional atomic coordinates for the unit cells of Zn-TAPB-COF-1, ZnZn-TAPB-COF-1 and Co-TAPB-COF-1 are summarised in Tables S2–S4. Notably, all M(salen)-COFs obtained via one-step synthesis exhibited considerably stronger diffraction peaks in PXRD than those obtained from two-step synthesis (Fig. S7)[26]. The stronger crystallinity of M(salen)-COFs from the one-step synthesis could be attributed to multicomponent co-existence, which provided a larger space for the reversible formation of imine bonds and structural self-healing of M(salen)-COFs[23,24,27]. Moreover, the possibility of scaling up this one-step synthesis for crystalline M(salen)-COFs was investigated. Crystalline Co-TAPB-COF-1 powder of 2.61 g with a single synthesis was also prepared from the one-step synthesis (Fig. S8) (Yield: 84%).

The chemical structures of the M(salen)-COFs were then verified using Fourier transform infrared (FT-IR) spectroscopy and solid-state nuclear magnetic resonance (NMR) spectroscopy. The FT-IR spectra showed typical bands for imine bonds (−C = N−) at 1608–1639 cm$^{-1}$ (Figs. S9–S12). Subsequently, $^{13}$C cross-polarisation magic angle spinning (CP/MAS) NMR spectra with characteristic carbon signals at 158–163 ppm further confirmed the formation of −C = N− in M(salen)-COFs (Figs. S13–S16). The peaks appearing around 65 ppm further confirmed the existence of C–N from ethylenediamine[4]. These results demonstrate that the M(salen)-COFs obtained from different synthesis methods possess the same chemical structures. Thermogravimetric analysis curves showed that all M(salen)-COFs could maintain thermal stability up to 300 °C (Fig. S17). $N_2$ adsorption−desorption isotherms at 77 K were used to assess the permanent porosity of M(salen)-COFs (Figs. S18–S23). For these M(salen)-COFs-1, all isotherms exhibited type IV(b) features, revealing that these COFs possessed mesopores. The

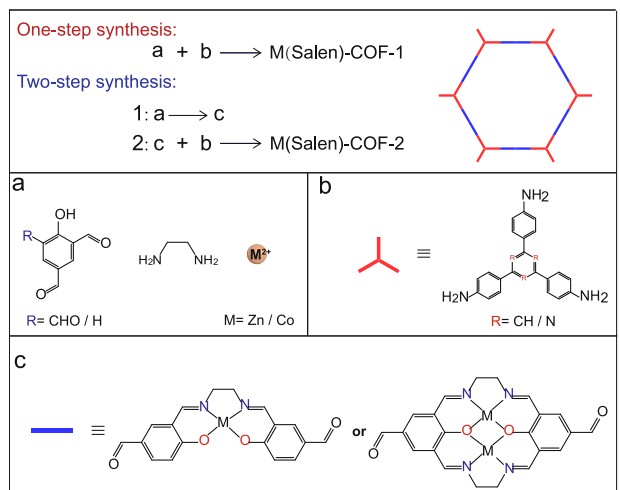

One-step synthesis:
a + b ⟶ M(Salen)-COF-1
Two-step synthesis:
1: a ⟶ c
2: c + b ⟶ M(Salen)-COF-2

a

R= CHO / H          M= Zn / Co

b

R= CH / N

c

**Fig. 1 | Schematic illustration of the synthesis strategies.** Synthesis strategies of metallosalen-covalent organic frameworks (M(salen)-COFs) (M = Zn and Co), including one-step synthesis (red route) and two-step synthesis (blue route). **a** Structures of 4-hydroxyisophthalaldehyde, ethylenediamine and metal species. **b** Structure of tridentate amine ligands. **c** Structures of M(salen)-CHO or MM(salen)-CHO.

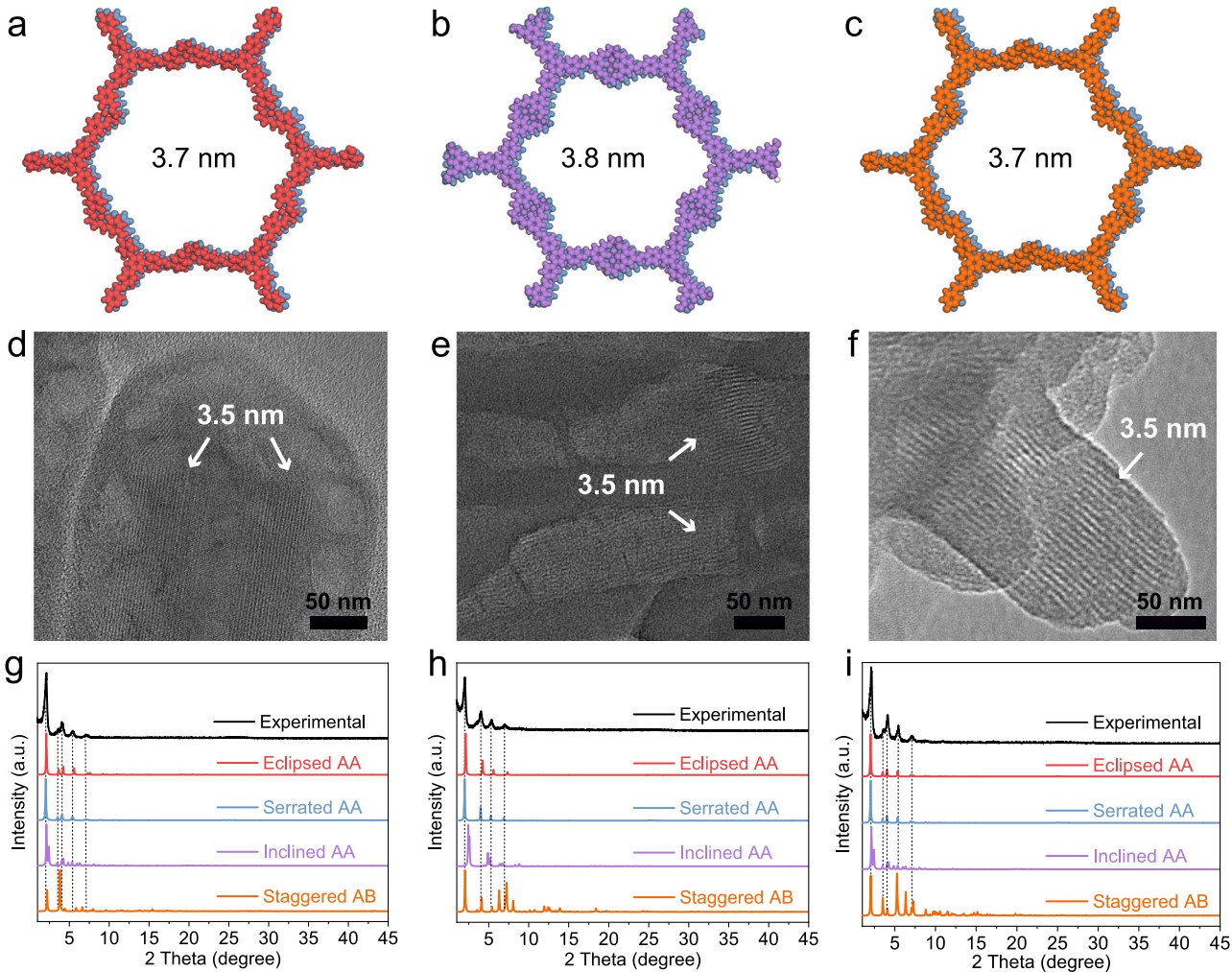

**Fig. 2 | Structure characterisation of M(salen)-COFs. a−c** Serrated AA-stacking structures, (**d−f**) TEM images and (**g−i**) experimental and simulated powder X-ray diffraction patterns of Zn-TAPB-COF-1, ZnZn-TAPB-COF-1 and Co-TAPT-COF-1, respectively.

Brunauer−Emmett−Teller surface areas ($S_{BET}$) of these M(salen)-COFs were distributed from 320 to 754 m² g⁻¹ (Table S5). Non-local density functional theory (NLDFT) calculations demonstrated that the M(salen)-COFs prepared using the one-step synthesis process possessed mesopores with a diameter of 3.5 nm[20]. Micropores (1.1 nm) and mesopores (3.8 nm) co-existed in M(salen)-COF-2 synthesised by the two-step synthesis[28]. The differences in pore sizes between M(salen)-COFs from one-step and two-step synthesis might be caused by partial pore collapse and lower crystallinity in M(salen)-COFs-2[29]. The theoretical pore size distributions of Zn-TAPB-COF-1 (~3.7 nm) and ZnZn-TAPB-COF-1 (~3.8 nm) were calculated using the Zeo + + programme (Figs. S5, 6)[30]. The experimental pore sizes maintained good consistency with the simulated values in serrated AA-stacking structures.

To study metal dispersion, electronic properties and local coordination structures in M(salen)-COFs, aberration-corrected transmission electron microscopy (AC-TEM), X-ray absorption near-edge structure (XANES) and extended X-ray absorption fine structure of synchrotron radiation (EXAFS) were conducted. AC-TEM images of Zn-TAPB-COF-1, ZnZn-TAPB-COF-1, and Co-TAPT-COF-1 all exhibited high-density bright dots without agglomerates, indicating their uniform isolated metal distribution (Fig. 3a−c)[31,32]. The Zn contents in Zn-TAPB-COF-1 and ZnZn-TAPB-COF-1, as detected via inductively coupled plasma atomic emission spectroscopy (ICP), were 8.23 wt% and 8.31 wt%, respectively. Co contents in Co-TAPB-COF-1 and Co-TAPT-COF-1 were 9.47 wt% and 10.34 wt%, respectively. Such high

metal loading with high dispersion was rare because aggregation to clusters or nanoparticles always occurred easily at metal contents above 2 wt%[33]. Abundant and highly dispersed metal species in these M(salen)-COFs were expected to exhibit excellent catalytic activities. Further, XANES and X-ray photoelectron spectroscopy (XPS) were used to confirm the electronic state of the metals in M(salen)-COFs (Figs. S24, 25). In the XANES spectra, the Zn K-edge of Zn-TAPB-COF-1 and ZnZn-TAPB-COF-1 was between those of Zn foil and ZnO, indicating that the oxidation states of Zn are 0 ~ + 2 (Fig. S24a)[34]. The Co K-edge of Co-TAPT-COF-1 was between the Co foil and CoO (Fig. S24b), indicating that the oxidation state of the Co species in Co-TAPT-COF-1 was also 0 ~ + 2. The XPS results indicated that the Co species in Co-COFs were +2 (Fig. S25). Subsequently, Fourier transformed (FT) EXAFS was used to reveal the local coordination structures of the metal atoms (Fig. 3d−f and Table S6). The FT-EXAFS spectrum of the Zn foil showed a major peak at 2.64 Å (with phase correction), which was assigned to Zn−Zn bonds. Meanwhile, ZnO exhibited two main peaks at 1.95 Å and 3.25 Å assigning to Zn-O and Zn-Zn bonds, respectively. FT-EXAFS spectra of Zn-TAPB-COF-1 and ZnZn-TAPB-COF-1 showed main peaks around 1.94 Å, corresponding to Zn−O or Zn−N bond lengths (Fig. 3d, e)[31,33]. No signature of Zn−Zn bonds was observed in the Zn-COFs above, thus validating the absence of Zn nanoparticles or clusters. Furthermore, EXAFS fitting results confirmed that the Zn atom was coordinated with approximately two N atoms and two O atoms in both Zn-TAPB-COF-1 and

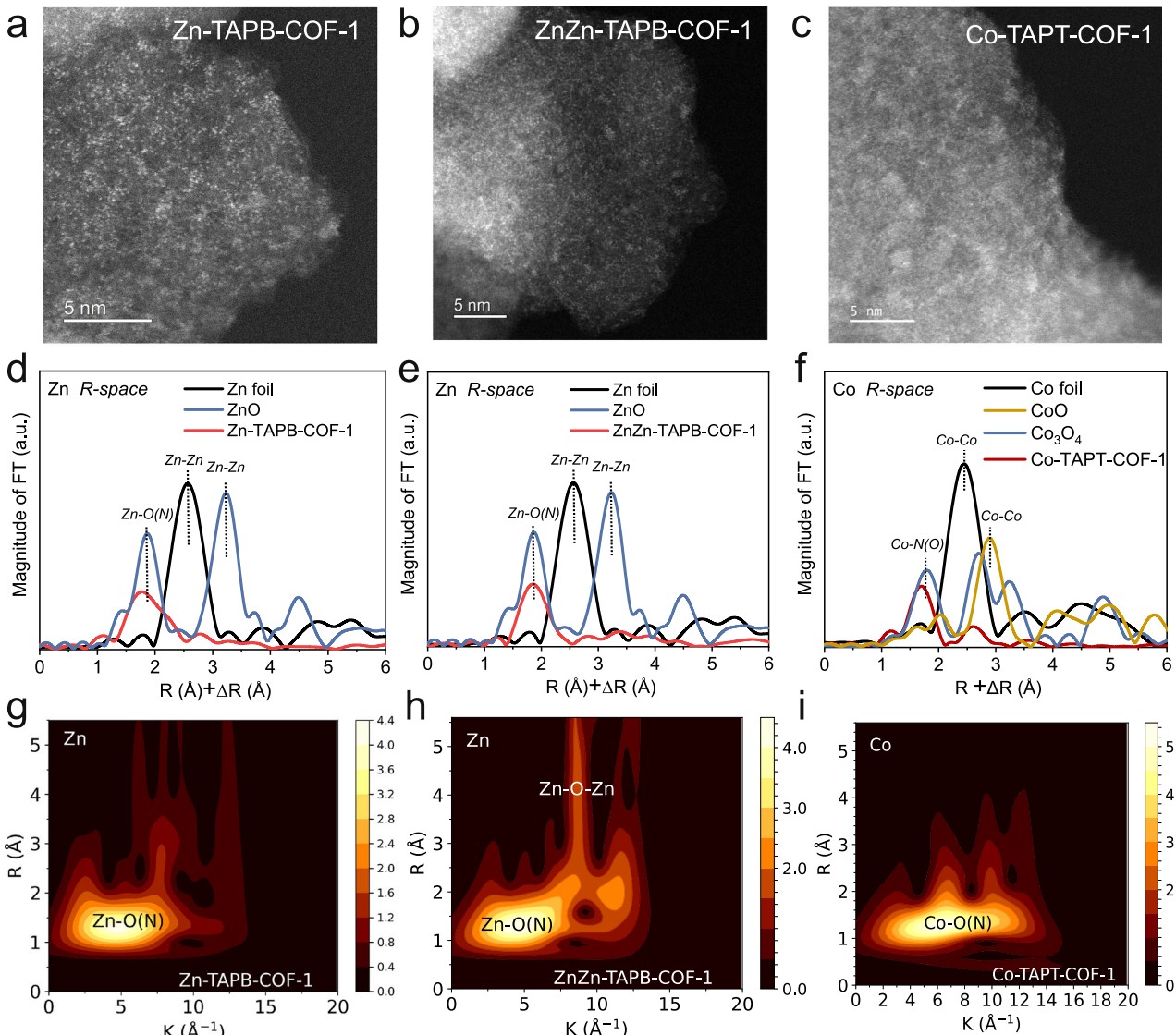

**Fig. 3 | Metal dispersion and coordination environment analysis.** Aberration-corrected transmission electron microscopy images for (**a**) Zn-TAPB-COF-1, (**b**) ZnZn-TAPB-COF-1 and (**c**) Co-TAPT-COF-1 respectively. **d** FT-EXAFS spectra for Zn *R*-space of Zn foil, ZnO and Zn-TAPB-COF-1. **e** FT-EXAFS spectra for Zn *R*-space of Zn foil, ZnO and ZnZn-TAPB-COF-1. **f** FT-EXAFS spectra for Co *R*-space of Co foil, CoO, $Co_3O_4$ and Co-TAPT-COF-1. **g–i** Wavelet transform (WT) for Zn-TAPB-COF-1, ZnZn-TAPB-COF-1 and Co-TAPT-COF-1, respectively.

ZnZn-TAPB-COF-1 (Table S6). FT-EXAFS spectra of Co foil, CoO and $Co_3O_4$ showed major peaks around 2.48 Å, 2.99 Å and 2.86 Å, corresponding to Co–Co and Co–O bonds (Fig. 3f). As for Co-TAPT-COF-1, two main peaks at 1.86 Å and 2.61 Å were assigned to Co–O (N) bonds from the salen moiety and Co–O bonds from $H_2O$. Moreover, the coordination number of Co atoms in Co-TAPT-COF-1 was six (Fig. 3f and Table S6), suggesting the existence of two O atoms from $H_2O$ on the axial positions except for the O/N atoms from the salen moiety[34–36]. Wavelet transform (WT) further visualised the metal species' coordination forms in M(salen)-COFs (Fig. 3g–i). The WT maximum (without phase correction) at 4.5 Å⁻¹ (*x*-axis: 1.5 Å) for Zn-TAPB-COF-1 could be assigned to the Zn–O or Zn–N bonds. As for ZnZn-COF-TAPB-1, there were two apparent shells in 4.5 Å⁻¹ (*x*-axis: 1.5 Å) and 9 Å⁻¹ (*x*-axis: 4.5 Å), which could be assigned to Zn–O (N) and Zn–O–Zn bonds. For Co-TAPT-COF-1, the WT maximum at 6.36 Å⁻¹ (*x*-axis: 1.43 Å) could be assigned to the Co–O or Co–N bonds (Fig. 3i). These characterisations led to the solid conclusion that the metal species were atomically dispersed across the whole range of M(salen)-COFs. Moreover, the coordination structures of metal atoms in Zn-TAPB-COF-1, ZnZn-TAPB-COF-1, and Co-TAPT-COF-1 were identified as $Zn–N_2O_2$, $Zn–N_2O_2–Zn–N_2O_2$ and $Co–N_2O_2$, respectively.

Thus far, precise construction of M(salen)-COFs superstructures without templates remains unclear[37]. The influence of synthetic routes on M(salen)-COF morphology has rarely been studied. Considering that the synthesis strategies mentioned above have considerable impact on the crystallinity of M(salen)-COFs, we hypothesise that these strategies might also affect the morphologies of M(salen)-COFs. As shown in Figs. S26–S33, M(salen)-COFs synthesised using different approaches exhibited different morphologies. For example, Zn-TAPB-COF-1 exhibited apparent nanobelts (Fig. S26), which differed from the hollow spheres of Zn-TAPB-COF-2 (Fig. S27). Most notably, Co-TAPT-COF-1 (Fig. S32), obtained by the one-step reaction, self-assembled into hollow nanotubes with a macropore channel (inner diameter of 135 nm and outer diameter of 220 nm) and a thin tube wall. These nanotubes all exhibited smooth surfaces without irregularities. In contrast, the Co-TAPT-COF-2 synthesised using the two-step method showed irregular nanosheets with slight curls (Fig. S33). Such nanotubes can

dramatically benefit photogenerated carriers' migration and mass transfer in photocatalysis. Moreover, the nanotube superstructure could benefit the $CO_2$ absorption ability (Fig. S34). According to reported work, distinct nanotubes may originate from the dynamic formation of imine bonds between unreacted aldehyde and amino groups[37]. The different morphologies of M(salen)-COFs resulted from differing reaction environments of the two syntheses strategies, which could profoundly affect the formation of imine bonds, crystal nucleation, and growth[38]. Fragments in the one-step synthesis tended to self-assemble into M(salen)-COFs with larger scales because of the co-existence of all raw materials. In the stepwise synthesis, 4-hydroxyisophthalaldehyde (or 2-hydroxybenzene-1, 3, 5-tri-carbaldehyde) and ethylenediamine were not present in the second step. Thus, condensation reactions in the second step occurred only on the highly dispersed nuclei generated from the first step, limiting the larger-scale growth of M(salen)-COFs[34,35]. In addition to the synthesis strategies, coordination environments, ligands and metal species affected the morphology of M(salen)-COF superstructures.

## Solar syngas production from $CO_2$

In recent years, photocatalytic $CO_2$ reduction to syngas (the mixture of CO and $H_2$ that can be converted into a variety of fuels) has become a hot research trend owing to its mild reaction conditions, high safety level and environmental friendliness[39,40]. However, the synthesis of syngas with controlled proportions for different downstream products remains a great challenge, hindering this method' implementation[41]. Notably, the structure–performance relationships of photocatalysts remain unrevealed. Here, we demonstrated highly tunable production of syngas with synthesised M(salen)-COFs as photocatalysts (Fig. 4a). Zn-TAPB-COFs, ZnZn-TAPB-COFs, Co-TAPB-COFs and Co-TAPT-COFs were selected as four typical photocatalysts to illustrate the effects of synthetic strategies, coordination environments, metal species and ligands on their photocatalytic performances under visible-light irradiation (Fig. 4, Figs. S35–S39 and Tables S7, S8). According to photocatalytic tests, Zn-TAPB-COF and ZnZn-TAPB-COF with Zn as metal centres both showed high $H_2$ production and low CO production rates (Fig. 4b, c). Co-TAPB-COFs and Co-TAPT-COFs with Co as metal centres all showed exceedingly high photocatalytic activities for both $H_2$ and CO production. In particular, Co-TAPT-COF-1 nanotubes exhibited a CO production rate of 8.39 mmol $g^{-1}$ $h^{-1}$ (83.92 μmol $h^{-1}$) and $H_2$ rate of 11.31 mmol $g^{-1}$ $h^{-1}$ (113.10 μmol $h^{-1}$) in 4 h (Fig. 4e and S34), which is comparable to the best reported COFs (Table S9)[42]. In addition, the quantum yield after 4 h was calculated to be 0.006%. The calculated turnover numbers for CO and $H_2$ were respectively 19.50 and 24.86 in 4 h. The calculated turnover frequencies for CO and $H_2$ were respectively 4.88 $h^{-1}$ and 6.22 $h^{-1}$ in 4 h. Here, M(salen) centres in M(salen)-COFs were expected to act as excellent $CO_2$ adsorption and activation sites[43,44]. Notably, Co-TAPT-COF-2 exhibited a CO production rate of 3.50 mmol $g^{-1}$ $h^{-1}$ and a $H_2$ production rate of 4.72 mmol $g^{-1}$ $h^{-1}$, revealing that the synthesis routes of M(salen)-COFs considerably affected their photocatalytic activity (Fig. 4e, f). The considerably higher activity of Co-TAPT-COF-1 than that of Co-TAPT-COF-2 might result from its high crystallinity and nanotube superstructure via one-step synthesis. Moreover, M(salen)-COFs ligands also influenced their photocatalytic performances. For example, Co-TAPB-COF-1 showed considerably lower syngas production rate (CO rate: 6.44 mmol $g^{-1}$ $h^{-1}$, $H_2$ rate: 6.71 mmol $g^{-1}$ $h^{-1}$) than Co-TAPT-COF-1 (Fig. 4d, e). With these photocatalysts, the proportion of $H_2$/CO could be continuously adjustable within 1:1 ~ 30:1 (Table S8). These results demonstrate that the synthesis routes, metal species, metal coordination environments, and ligands in M(salen)-COFs play important roles in regulating photocatalytic activity[45]. Subsequently, a series of control experiments were conducted to identify the key factors for $CO_2$-to-CO conversion (Fig. 4g). In the absence of Co-TAPT-COF-1, the photosensitizer (Ru complex) produced a small amount of CO and $H_2$. No CO was detected

without $CO_2$ and light irradiation, proving that CO originated from photocatalytic $CO_2$ reduction. To further determine the carbon source of CO, isotope labelled carbon dioxide ($^{13}CO_2$) was employed as the source gas (Fig. 4h and S40). The mass spectrometry spectrum showed a $^{13}CO$ ($m/z = 29$) signal when using $^{13}CO_2$ as the source gas. No $^{12}CO$ ($m/z = 28$) was detected, demonstrating that all the CO originated from $CO_2$[46,47]. To assess the stability of Co-TAPT-COF-1, recycling experiments and post-catalysis characterisations, including PXRD, FT-IR and $N_2$ adsorption–desorption isotherms, were further conducted (Figs. S41, S42). As shown in Fig. S41, after continuous tests for five cycles, Co-TAPT-COF-1 retained photocatalytic activity comparable to that of the initial tests. Moreover, the PXRD patterns of Co-TAPT-COF-1 after catalysis retained sharp peaks (Fig. S42a). According to FT-IR spectra in Fig. S42b, the signal for the −C=N− linkages in Co-TAPT-COF-1 could still be detected at 1621 $cm^{-1}$. These results demonstrate the excellent stability of Co-TAPT-COF-1 in photocatalysis. In addition, despite minor changes in the BET surface area values, the pore diameter for Co-TAPT-COF-1 was still ~3.5 nm (Fig. S42c, d).

To reveal the roles of metal species and coordination environments of M(salen)-COFs in photocatalysis, density functional theory (DFT) calculations and spectroscopic characterisations were performed. First, free energies for the conversion of $CO_2$ to CO on model catalysts, including M(salen)-TAPT (M = Zn, ZnZn and Co), were calculated by DFT calculations (Fig. S43). According to the results, the formation of $*CO_2H$ species on M(salen)-TAPT was the rate-determining step (RDS) for CO production (Fig. 4i). More particularly, Co-TAPT-COF-1 exhibited the lowest RDS free energy (0.75 eV), which was consistent with the best photocatalytic performance of Co-COFs compared to other M-TAPT-COFs (2.25 eV and 2.02 eV for Zn-TAPT-COF and ZnZn-TAPT-COF; Table S10). When compared with scholarly references, the RDS free energy on Co-TAPT-COF was considerably lower than that of the Ni−$N_4$ site in NiPc-Co-POP, but higher than that of the Co−$N_4$ site in COF-367 NSs, consistent with their photocatalytic activities[41,45]. Further, spectroscopic characterisations of M(salen)-COFs were conducted to investigate their photoelectric properties. The UV−vis diffuse reflectance spectra revealed that all Co(salen)-COFs exhibited broad visible light absorption (Fig. S44a). The band gaps (Eg) of Co-TAPT-COF-1 and Co-TAPT-COF-2 were nearly the same (2.38 eV and 2.33 eV, respectively; Fig. S44). Mott−Schottky analysis was conducted to estimate the relative conduction band minimum (CBM) positions of Co(salen)-COFs (Fig. S45). The positive slopes of the curves indicated their n-type semiconductor characteristics, in which electrons acted as majority carriers[48]. Furthermore, ultraviolet photoelectron spectroscopy was conducted to confirm their valence band maximum (VBM) positions (Fig. S46). In addition, the band structures of the Co-COFs could be calculated from the aforementioned characterisations (Table S11 and Fig. S47), indicating that all Co-COFs were suitable for photocatalytic $CO_2$ reduction.

## Discussion

In conclusion, the one-step synthesis strategy proposed in this work provided a facile and scalable method for preparing M(salen)-COFs (M = Zn, ZnZn, and Co). M(salen)-COFs with different metal coordination environments (M−$N_2O_2$ and M−$N_2O_2$−M−$N_2O_2$) were fabricated and the structures were confirmed by XAFS spectra. The M(salen)-COFs obtained from this one-step strategy showed considerably higher crystallinity and larger-scale superstructures than those obtained from the two-step synthesis, exhibiting superior photocatalytic performance. In particular, Co-TAPT-COF-1 exhibited distinct nanotube superstructures and showed the highest activity towards syngas production from $CO_2$ photoreduction (CO and $H_2$ production rates of 8.39 mmol $g^{-1}$ $h^{-1}$ and 11.31 mmol $g^{-1}$ $h^{-1}$, respectively). The ratio of n($H_2$)/n(CO) was widely tunable by regulating the M(salen)-COF structures and synthesis routes. These results provided insights into the structure–property relationships and design principles of high-

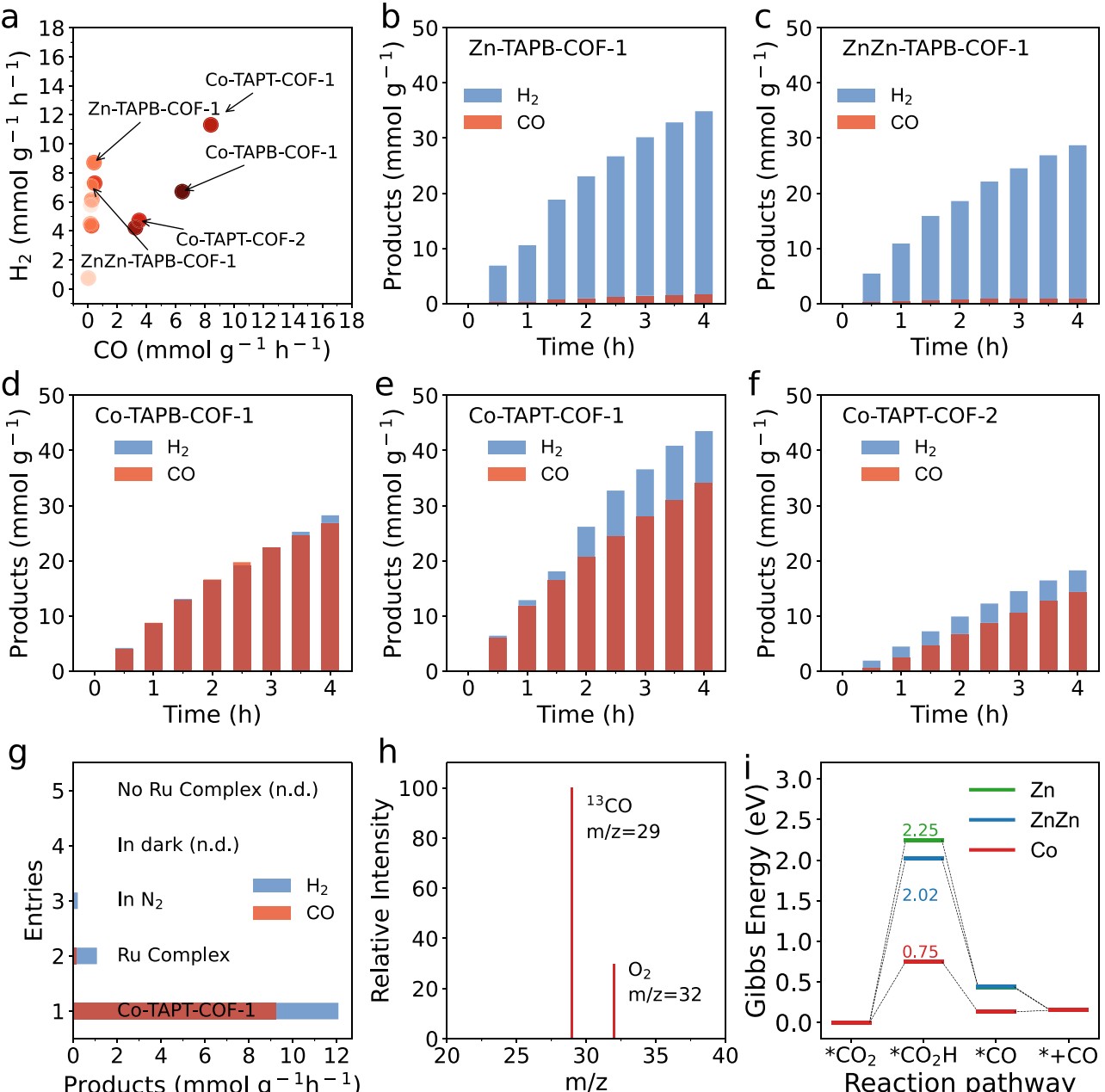

**Fig. 4 | Photocatalytic conversion of CO₂ to syngas. a** Syngas production rate with M(salen)-COFs synthesised in this work. Photocatalytic syngas production with (**b**) Zn-TAPB-COF-1, (**c**) ZnZn-TAPB-COF-1, (**d**) Co-TAPB-COF-1, (**e**) Co-TAPT-COF-1 and (**f**) Co-TAPT-COF-2. **g** Control experiments under different reaction conditions. **h** Mass spectroscopy of the product ($^{13}CO$: $m/z = 29$) in the photocatalytic reduction of $^{13}CO_2$ with Co-TAPT-COF-1 as the photocatalyst. **i** Calculated free energy diagrams for $CO_2$ reduction to CO on Zn-TAPT-COF, ZnZn-TAPT-COF and Co-TAPT-COF.

efficiency photocatalysts. Moreover, the high yield and mild conditions of this one-step synthesis laid the foundation for exploring the industrial applications of M(salen)-COFs.

## Methods

### Synthesis of M(salen)-CHO (M = Zn and Co)

4-hydroxyisophthalaldehyde (5 mmol; 750.66 mg) and Zn(OAc)₂·2H₂O (2.5 mmol; 548.78 mg) were added to a 50-mL two-necked flask. Further, degassed ethanol (15 mL) and ethylenediamine (2.5 mmol; 0.16 mL) were added to the flask. The mixture was stirred at 50 °C under N₂ atmosphere for 5 h. Upon cooling, Zn-CHO was obtained by washing with cold methanol three times (yield: 90%). Co-CHO was synthesised similarly to Zn-CHO except that Zn(OAc)₂·2H₂O was replaced with Co(OAc)₂·4H₂O (yield: 84%).

### Synthesis of ZnZn-CHO

2-hydroxybenzene-1, 3, 5-tricarbaldehyde (5 mmol; 890.71 mg) and Zn(OAc)₂·2H₂O (5 mmol; 1.097 g) were added to a 50-mL two-necked flask. Further, degassed ethanol (20 mL) and ethylenediamine (5 mmol; 0.33 mL) were added to the flask. The mixture was stirred at 50 °C under a N₂ atmosphere for 5 h. Upon cooling, ZnZn-CHO was obtained by washing with cold methanol three times (yield: 53%).

### Synthesis of M(salen)-COFs (M = Zn and Co)

In a one-step synthesis of Zn-TAPB-COF-1, 4-hydroxyisophthalaldehyde (0.9 mmol; 135 mg) and Zn(OAc)₂·2H₂O (0.45 mmol; 98.78 mg) were added into a 20-mL Teflon-sealed autoclave. Further, 4 mL of ethanol and ethylenediamine (0.45 mmol; 30 μL) were added. After sonication for 5 min, 2 mL of o-DCB, 0.5 mL of DMF, and 1.6 mL of

HOAc (3 M) were added to the above suspension. After another 5 min sonication, 1, 3, 5-tri(4-aminophenyl) benzene (TAPB; 0.3 mmol; 105.44 mg) was added to the reaction system. The resulting mixture was sonicated for 15 min and heated at 120 °C for 72 h. Finally, yellow powder was obtained after washing with ethanol and THF three times. The yellow powder was soaked in methanol and then in liquid $CO_2$ (7.5 MPa, 25 °C) for 24 h. Finally, the powder was dried using super-critical $CO_2$ (scCO$_2$, 8 MPa, 35 °C) treatment for 24 h (yield: 87%). Additionally, Zn-TAPT-COF-1 was synthesised similarly to Zn-TAPB-COF-1, except that TAPB was replaced with TAPT (yield: 67%). Co-TAPB-COF-1 was synthesised similarly to Zn-TAPB-COF-1, except that Zn(OAc)$_2$·2H$_2$O was replaced with Co(OAc)$_2$·4H$_2$O (yield: 83%).

In a two-step synthesis of Zn-TAPB-COF-2, Zn-CHO was first synthesised. Further, Zn-CHO (0.15 mmol; 60 mg) and TAPB (0.2 mmol; 70 mg) were added to a 20-mL Teflon-sealed autoclave. Subsequently, 6.4 mL of n-BuOH, 1.6 mL of o-DCB, and 0.3 mL of HOAc (3 M) were added to the reaction system. The autoclave was then heated in an oven at 120 °C for 72 h, and yellow Zn-TAPB-COF-2 powder was obtained after being washed with ethanol and THF three times. The powder was soaked in methanol and then in liquid CO$_2$ (7.5 MPa, 25 °C) for 24 h. Finally, the powder was dried using scCO$_2$ (8 MPa, 35 °C) treatment for 24 h (yield: 67%). Zn-TAPT-COF-2 was synthesised similarly to Zn-TAPB-COF-2, except that TAPB was replaced with TAPT (yield: 66%). Similarly, Co-TAPB-COF-2 was synthesised in a similar way with Zn-TAPB-COF-2, except that Zn-CHO was replaced by Co-CHO (yield: 53%).

### Synthesis of ZnZn-COFs

For one-step synthesis of ZnZn-TAPB-COF-1 sample, 2-hydro-xybenzene-1, 3, 5-tricarbaldehyde (0.9 mmol; 160.33 mg) and Zn(OAc)$_2$·2H$_2$O (0.45 mmol; 98.78 mg) were added to a 20-mL Teflon-sealed autoclave. Further, 4 mL of ethanol and ethylenediamine (0.45 mmol; 30 μL) were added. Following sonication for 5 min, 2 mL of o-DCB and 0.5 mL of DMF and 1.6 mL of HOAc (3 M) were added to the suspension. After sonication for another 10 min, 1, 3, 5-tri(4-amino-phenyl) benzene (TAPB; 0.3 mmol; 105.44 mg) was added to the reaction system. The obtained suspension was sonicated for 15 min and then heated at 120 °C for 72 h. Yellow–brown powder was obtained after washing with ethanol and THF three times. The powder was soaked in methanol and then in liquid CO$_2$ (7.5 MPa, 25 °C) for 24 h. Finally, scCO$_2$ (8 MPa, 35 °C) treatment for 24 h was used to dry the powder (yield: 81%). In addition, ZnZn-TAPT-COF-1 was synthesised similarly to ZnZn-TAPB-COF-1, except that TAPB was replaced with TAPT (yield: 75%).

For two-step synthesis of ZnZn-TAPB-COF-2, ZnZn-CHO was first synthesised. Further, ZnZn-CHO (0.3 mmol; 120 mg) and TAPB (0.2 mmol; 70 mg) were added to a 20-mL Teflon-sealed autoclave. Moreover, 6.4 mL of n-BuOH, 1.6 mL of o-DCB, and 0.7 mL of HOAc (3 M) were added to the reaction system, which was then heated at 120 °C for 72 h. Yellow-brown ZnZn-TAPB-COF-2 powder was obtained after washing with ethanol and THF three times. The powder was soaked in methanol and then in liquid CO$_2$ (7.5 MPa, 25 °C). Finally, scCO$_2$ (8 MPa, 35 °C) treatment for another 24 h was used to dry the powder (yield: 39%). In addition, ZnZn-TAPT-COF-2 was synthesised similarly to ZnZn-TAPB-COF-2, except that TAPB was replaced with TAPT (yield: 36%).

### Characterisation

Powder X-ray diffraction was performed on a Rigaku D-MAX 2500/ PC. FT-IR spectra were performed on a Bruker VERTEX 70 V. $^{13}$C solid-state spectra were recorded using a Varian Infinity-400 spectro-meter. TGA was performed using a Netzsch Model STA 449 C micro-analyser under N$_2$ atmosphere from 25 °C to 900 °C. N$_2$ absorption and desorption isotherms were obtained using a quantachrome automated surface area & pore size analyzer, and the corresponding

pore size distributions were estimated through nonlocal density functional theory (NLDFT). XPS spectra were acquired from an ESCALAB 250XI with a monochromatic Al Kα X-ray source. Products in a $^{13}$C labelled photocatalytic experiment were detected using a SHIMADZU GC-MS-QP2010 SE equipped with a ShiCap-PLOT Q column.

### Solar syngas production from CO$_2$

Photocatalytic CO$_2$ reduction was performed using the Labsolar-6A system (Beijing Perfectlight Technology Co., Ltd.) connected using Agilent 8890 gas chromatography. In a typical experiment, M(salen)-COFs (10 mg) were dispersed in a H$_2$O/ TEOA/ CH$_3$CN mixture (1:2:7; 100 mL) and then [Ru(bpy)$_3$]Cl$_2$·6H$_2$O (60 mg) was added. Further, the mixture was thoroughly degassed and then backfilled with pure CO$_2$ to a pressure of 75 KPa three times. A 300 W xenon lamp with a UV cut-off filter (λ ≥ 420 nm) was used as the light source. During the photocatalytic process, the mixture was vigorously stirred with a magnetic stirrer (400 rpm) and cooled via circulating water to maintain a temperature of approximately 25 °C. The influence of [Ru(bpy)$_3$]Cl$_2$·6H$_2$O contents was tested with Co-TAPT-COF-1 (10 mg) under the same solvent system as above. Dif-ferent TEOA and H$_2$O proportions (2:1, 3:1 and 4:1) were also investigated using Co-TAPT-COF-1 (10 mg) and [Ru(bpy)$_3$]Cl$_2$·6H$_2$O (60 mg), respectively. To conduct the recirculation test, Co-TAPT-COF-1 (10 mg) and [Ru(bpy)$_3$]Cl$_2$·6H$_2$O (60 mg) were dispersed in a H$_2$O/TEOA/CH$_3$CN mixture (1:2:7, 100 mL). After photocatalytic testing for 4 h, the sample was filtered through an organic filter membrane and used for subsequent recycling tests. For the AQE test, Co-TAPT-COF-1 (17 mg) and [Ru(bpy)$_3$]Cl$_2$·6H$_2$O (60 mg) were dispersed in a H$_2$O/TEOA/CH$_3$CN mixture (1:2:7; 100 mL). Further, the mixture was thoroughly degassed and then backfilled with pure CO$_2$ to 75 KPa three times. A 50 mW cm$^{-2}$ xenon lamp with a UV band-pass filter (λ = 420 nm) was used as the light source.

### DFT calculations

In the DFT calculations, three model catalysts, including M(salen)-TAPT (M = Zn, ZnZn, and Co), were used as calculated models. Geometry optimisations and frequency calculations were conducted using the B3LYP functional[49,50]. The van der Waals correction of Grimme's D3 scheme[51,52] was incorporated to describe noncovalent interactions between the adsorbent and the adsorbate. The 6-31 G (d) basis set was used for the nonmetal atoms (C, H, O and N), and the SDD effective core potential was used for the metal atoms. For each geometric stationary point, single-point energy calculations were performed using an extended 6-31 + + G(d,p) basis set for the non-metal atoms and the SMD solvation model was used to determine the effect of the solvent (water)[49,53]. Thermal corrections to the free energy of each reactant, product and intermediate state were cal-culated via frequency calculations (298.15 K and 1 atm) based on optimised geometry. All DFT calculations were performed using Gaussian 16 software[54].

The reactant and product states in the photocatalytic CO$_2$ reduction process, as well as in intermediate states were proposed and evaluated. The reaction scheme is as follows:

$$I : \ ^* + CO_2 + H^+ + e^- \rightarrow {}^*CO_2H \tag{S1}$$

$$II : \ ^*CO_2H + H^+ + e^- \rightarrow {}^*CO + H_2O \tag{S2}$$

$$III : \ ^*CO \rightarrow {}^* + CO \tag{S3}$$

Here, * refers to the active site on computational models. The free energy of (H$^+$ + e$^-$) at standard conditions was assumed to be the energy of 1/2H$_2$. The free energy change is described by the following

expressions:

$$\Delta G_I = G_{*CO_2H} - G_{CO_2} - 0.5 G_{H_2} - G_* \tag{S4}$$

$$\Delta G_{II} = G_{*CO} + G_{H_2O} - G_{*CO_2H} - 0.5 G_{H_2} \tag{S5}$$

$$\Delta G_{III} = G_{CO} + G_* - G^*_{CO} \tag{S6}$$

## Data availability

Source data are provided with this paper. The data that support the findings of this study are also available from the corresponding author [C.C.L. and W.Q.D.] upon reasonable request. Source data are provided with this paper.

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

## Acknowledgements

This work was supported by the National Key Research and Development Program of China (No. 2022YFA1503100 to W.Q.D.), National Natural Science Foundation of China (No. 22003031 to C.C.L., No. 22273050 to L.S.), Natural Science Foundation of Shandong Province (No. ZR2020QB080 to C.C.L., No. ZR2022MB098 to L.S.), Fundamental Research Funds of Shandong University (No. 2019GN109 to C.C.L.) and the Program of Young Scholars Future Program of Shandong University.

## Author contributions

W.Q.D. and C.C.L. planned and designed the project. W.Z. designed and synthesised the COFs. W.Z., X.W., N.J.L. and G.Q.R. completed characterisation and analysis, including TEM, XANES, XAFS and XPS analysis. W.Z., W.L.Z. D.C. and Z. L. performed the photocatalytic tests. P.G.H conducted nanosecond transient absorption spectra. W.Z. and L. S. executed DFT calculations. All authors were involved in the writing of the manuscript.

## Competing interests

The authors declare no competing interests.
