## [Peer Review File · Nature Communications]

Photocatalytic CO₂ reduction to syngas using metallosalen covalent organic frameworksREVIEWER COMMENTS

Reviewer #1 (Remarks to the Author):

This manuscript reports a one-step synthesis strategy for highly crystalline M-COFs without vacuum evacuation. M-COFs possessing controllable coordination environments of single-atomic metal sites and dual-atomic metal sites can be obtained, acting as photocatalysts for tunable syngas production from CO₂. It was characterized by SEM, AC-TEM, PXRD, FT-IR, XAFS, NMR and XPS et al. However, the relationship between the structure of the materials synthesized in the manuscript and the synthesis strategy is not clear. Whether materials with similar structures can be synthesized under the same conditions, the paper does not see its repetitive support materials. However, the manuscript contains many problems, and your findings need to be significantly revised in combination with the points I mentioned below. The revised version can be published in Nature Communications.

1. Whether the three metal complexes in Fig. 1b can also generate MOFs, Fig. 2 only shows the structural model (a-c), experiment (red) and Pawley refined (black) PXRD pattern (d-f) of Zn-TAPB-COF-1, ZnZn-TAPB-COF-1 and Co-TAPB-COF-1. It is suggested to provide PXRD diagrams of MOFs for comparison, indicating that M-COFs are generated. At the same time, explain what software is used to simulate the parallel AA-stacking structure, supplement the misplaced AB-stacking structure for comparison, and indicate that the AA stack is not the AB stack. It is recommended that the experimental line and analog line in the PXRD diagram should be separated and not overlapped so that the black line cannot be seen clearly.
2. The regularity of BET data listed in Table S3 is not good,, such as Zn-TAPB-COF-1 836 m²g⁻¹, Zn-TAPB-COF-2 338 m²g⁻¹, Zn-TAPT-COF-1 319 m²g⁻¹ and Zn-TAPT-COF-2 1616 m²g⁻¹. Since this paper does not study the influence of synthesis strategy on the morphology and structure of M-COF, and the support of proving that the synthesis is M-COF is insufficient, how about the repeatability of one-step synthesis strategy? It is recommended to supplement supporting data.
3. Please explain why Zn-TAPT-COF-1 has the lowest specific surface area but the best catalytic performance in Table S3.
4. Why does the synthesis route greatly affect the photocatalytic activity?
5. The model for calculating the free energy of CO₂ conversion to CO is too simple. It is just a metal ligand, and does not include the covalent structure in COF. It is recommended to redesign the calculation model to explain where the active center of the catalyst is.
6. The three catalysts in Fig. S3a have good absorption between 400-600 nm. Why do you add photosensitizer [Ru (bpy) ₃] Cl₂ · 6H₂O.
7. It is suggested to supplement the circulating experimental data of the catalyst for CO₂ reduction to explain the chemical stability of the catalyst.

Reviewer #2 (Remarks to the Author):

This manuscript describes the synthesis of COFs for CO₂ reduction. One big problem is that the PXRD patterns of COFs are inconsistent with their simulated or proposed structures. Another key issue is the experimentally observed pore size is far different from the proposed structure. The 'one-pot' method has been reported previously for a similar coordination system and is not novel. The catalytic activity and selectivity of the COFs are not impressive compared to previously reported other framework catalysts. Eye-catching phrases 'Metallo-COFs' and 'single atom' should be used carefully and avoided; 'metallo-salen COFs' is more accurate for this case. I cannot suggest the publication in Nat Commun.

1. In figure 1c, X should be CH or N.
2. The synthesis routes indicate that in figure 1 are quite confusing. Could you use a clearer layout or arrows to better convey the differences between these two methods?
3. In the legends of figure 2d-f, the black curve should be "Pawley-refined" instead of "simulated".
4. Could you show the exact peak position of the PXRD patterns or provide cif files of the simulated

structures? The 2 θ positions of the major peaks seem to be a bit too high if the COF structures are actually as drawn in figure 2a-c.

5. In principle, the aldehyde groups can directly react with the tridentate amine ligands (TAPB/TAPT), which is a competition reaction of them reacting with ethylenediamine. If so, COF can also be produced. From the characterizations in this manuscript, it cannot rule out the possibility of these two reactions happening at the same time.

6. From the nitrogen sorption isotherms and pore size distributions, the pores of these M-COFs seem to be highly distributed, indicating the disordered structures.

7. The influence of synthetic methods is frequently emphasized in this manuscript. However, all the contrasts between M-COF-1 to M-COF-2 can be attributed to their crystallinity difference (as shown in Figure S5). Amorphous counterparts should be put into comparisons.

8. What is the stability of these COFs under photocatalytic conditions?

Reviewer #3 (Remarks to the Author):

This manuscript described one-step synthesis of three M-COFs, which were further used as photocatalysts. Characterizations have been made to analyze the M-COFs and to evaluate their performance for CO₂ photoreduction. The topic is interesting, but the authors failed to demonstrate enough novelty and impact which can demand the publication in Nature Communications. The monomers and reactions are well-known. The crystallinities of the products do not reach the state of the art in this field. In addition to some flaws and claims without clear support, there are critical concerns about the modelled structures of the M-COFs, i.e., the layer stacking. Therefore, I cannot recommend its publication in Nature Communications. The detailed comments and questions are listed as following.

1. In the abstract, the authors identified low crystallinities of COFs are the bottleneck of the field. While the statement is true, the high crystallinity of the COFs claimed by the authors in this study is misleading. The COFs only show less than ten peaks in the PXRD pattern, which is far from high crystallinity in the state of the art of M-COFs. See e.g., Nat. Commun. 2020, 11, 1434.
2. The unit cell parameters and space groups in Figure 2 are inconsistent to those in Tables S1 and S2 and in the text.
3. The Pawley fitting shown in Figure 2 have serious issues. (i) There is no space group of P6₂m. (ii) For Figure 1d, there should be more Bragg positions before 5 degree for the space group P-6₂m and the given unit cell. (iii) Why the c-axis of ZnZn-TAPB-COF-1 is 19.99 Å while the structure model looks having the same layer stacking as the others? (iv) Why there is no peak at the first several Bragg positions?
4. In modelling the structures of the COFs, the authors only considered the AA eclipsed stacking, which could cause misleading. How about other common layer stacking behaviors, such as AA inclined and AB staggered stacking?
5. To justify the structural models, I suggest to include a figure comparing experimental PXRD patterns with those calculated from the structural models.
6. How large are the pores measured from the structural models and how good do they agree with the N₂ sorption results? The pore size distributions in Figures S16-21 shows two pore sizes. One of c.a. 1.5 nm, and another of c.a. 3.0 nm. The authors need to describe where they come from.
7. The authors mentioned the yield of ZnZn-TATP-COF-1 on line 130, and conclude the one step synthesis as having high yield. This is misleading. The authors need to show the yield of other one step made COFs and compare the yields to two step synthesis.
8. On line 150, the authors claimed that one step synthesis show higher BET surface area than those synthesized by two steps. This is not true. Table S3 shows that half are the opposite.
9. The Zn-O bonding distance should be in the range of 1.9-2.1 Å for standard zinc oxides. The EXAFS results shown in Figure 3 show a distance of c.a. 1.6 Å, which has a significant deviation.
10. The section of morphological control of the COFs reads abruptly. Do the morphologies related to the focus of this manuscript, i.e., performance of CO₂ reduction?

11. It is difficult to access the structural models from Tables S1 and S2. I suggest to upload cif files as supplementary information, even though they are modelled structures.
12. The authors need to check the convention of writing space groups. For example, there is no space group should be written as P6/M.

Reviewer #4 (Remarks to the Author):

The synthesis of metallo-covalent organic frameworks (M-COFs) with high crystalline is highly desirable for their application in photocatalysis. This manuscript reported a very easy one-pot synthesis of highly crystalline M-COFs under ambient conditions and further demonstrated their application for syngas production from CO₂. The authors clearly characterized the structures of M-COFs with XRD, TEM, and EXAFS. Meanwhile, the authors demonstrated the generality of the methods for synthesizing other M-COFs. In photocatalysis, this M-COF disclosed a high syngas production rate. In overall, this work represents the new development in the synthesis and application of M-COFs. Considering the high novelty and significance, this Reviewer recommends the acceptance of this manuscript for publication in Nature Communications. There are some issues that should be addressed before publication.

1. Figure 1 presents the synthetic routes of M-COFs. However, it is not easy-to-understand. More text descriptions should be added in Figure 1.
2. In this manuscript, alicylaldehyde was conducted as the ligand for the synthesis of M-COFs. Did the authors check other ligands or metal species? This would be helpful for the readers to synthesize other M-COFs.
3. As the metal species in the M-COFs are all in single-atomic or dual-atomic state, the TOF of CO production should be given and compared with references.
4. How about the structural stability of the M-COFs? This would be very important for their application in catalysis.

Point-by-point Response to the Comments

Reviewer #1:

This manuscript reports a one-step synthesis strategy for highly crystalline M-COFs without vacuum evacuation. M-COFs possessing controllable coordination environments of single-atomic metal sites and dual-atomic metal sites can be obtained, acting as photocatalysts for tunable syngas production from CO₂. It was characterized by SEM, AC-TEM, PXRD, FT-IR, XAFS, NMR and XPS et al. However, the relationship between the structure of the materials synthesized in the manuscript and the synthesis strategy is not clear. Whether materials with similar structures can be synthesized under the same conditions, the paper does not see its repetitive support materials. However, the manuscript contains many problems, and your findings need to be significantly revised in combination with the points I mentioned below. The revised version can be published in Nature Communications.

Response: We thank Reviewer #1 for the positive comments. We have modified this manuscript based on your valuable comments and we believe that the revisions are much clearer. We have also synthesized M-COFs with nickel (Ni) salt via one-step synthesis. The synthesized Ni-TAPB-COF-1 also exhibited certain crystallinity (Figure R1).

Figure R1. PXRD pattern of Ni-TAPB-COF-1.

Comment 1: Whether the three metal complexes in Fig. 1b can also generate MOFs. Fig. 2 only shows the structural model (a-c), experiment (red) and Pawley refined (black) PXRD pattern (d-f) of Zn-TAPB-COF-1, ZnZn-TAPB-COF-1 and Co-TAPB-COF-1. It is suggested to provide PXRD diagrams of MOFs for comparison, indicating that M-COFs are generated. At the same time, explain what software is used to simulate the parallel AA-stacking structure, supplement the misplaced AB-stacking structure for comparison, and indicate that the AA stack is not the AB stack. It is recommended that the experimental line and analog line in the PXRD diagram should be separated and not overlapped so that the black line cannot be seen clearly.

Response: We thank the reviewer for this valuable comment. The key difference between MOFs and M-COFs is that metal atoms act as the indispensable nodes in MOFs, whereas metal atoms are dispensable in M-COFs. In this work, COFs are generated instead of MOFs based on the following two pieces of evidence. First, we synthesized the COFs without metal salts (N-TAPB-COF-1 and NN-TAPB-COF-1, which were counterparts to Zn-TAPB-COF-1 and ZnZn-TAPB-COF-1, respectively) and obtained crystal products with similar peak positions with their counterparts (Please see Figure R2). This proves that metal atoms are not indispensable nodes in the crystalline

products. Second, the peaks around 65 ppm in ^{13}C -NMR spectra of the M-COFs confirm the C-N from ethylenediamine, which act as necessary linker in the Salen-COFs (Please see Figures S13-S16) ^[R1-R2].

As suggested by the reviewer, we have provided the simulated PXRD patterns of the MOF counterparts to Zn-TAPB-COF-1 and ZnZn-TAPB-COF-1 (Figure R3), demonstrating that the products are M(salen)-COFs. Materials Studio is the software which is used to simulate the structures. We have also added the AB stacking structures for comparison (Please see Figure 2 in the revised Manuscript). The peak positions and relative intensities in experimental line all coincided better with AA stacking than AB stacking. Moreover, the pore size (~ 3.5 nm) detected by N_2 adsorption-desorption isotherms of the M(salen)-COFs were entirely consistent with AA stacking (3.7-3.8 nm) (Please see Figure 2 in the revised Manuscript, Figures S5-S6 and Figures S18-S23 in the revised Supporting Information). In addition, the experimental line and analog line in the PXRD diagram have been separated.

Figure R2. PXRD patterns of the (a) N-TAPB-COF-1 (counterpart to Zn-TAPB-COF-1) and (b) NN-TAPB-COF-1 (counterpart to ZnZn-TAPB-COF-1) synthesized without metal salts.

Figure R3. (a) Experimental PXRD pattern for Zn-TAPB-COF-1, simulated PXRD patterns for Zn-TAPB-COF-1 and Zn-TAPB-MOF-1. (b) Experimental PXRD pattern for ZnZn-TAPB-COF-1, simulated PXRD patterns for ZnZn-TAPB-COF-1 and ZnZn-TAPB-MOF-1.

Comment 2: The regularity of BET data listed in Table S3 is not good, such as Zn-TAPB-COF-1 $836 \text{ m}^2\text{g}^{-1}$, Zn-TAPB-COF-2 $338 \text{ m}^2\text{g}^{-1}$, Zn-TAPT-COF-1 $319 \text{ m}^2\text{g}^{-1}$ and Zn-TAPT-COF-2 $1616 \text{ m}^2\text{g}^{-1}$. Since this paper does not study the influence of synthesis strategy on the morphology and structure of M-COF, and the support of proving that the synthesis is M-COF is insufficient, how about the repeatability of one-step synthesis strategy? It is recommended to supplement supporting data.

Response: We thank the reviewer for pointing this out. We have retreated the COFs with solvent soaking and supercritical CO_2 (scCO_2) drying to remove the impurities in pores and maintain the mesopores of the COFs. The detailed descriptions have been added in the Methods section in the revised manuscript (Please see pages 17-19 in the revised Manuscript). After solvent soaking and scCO_2 drying, the regularity of BET data of M(salen)-COFs was greatly improved (Please see Table S5 and Figures S18-S23 in the revised Supporting Information). Moreover, the repeatability of one-step synthesis strategy was also characterized with Zn-TAPB-COF-1 as an example, showing similar crystallinity, BET surface area and pore volume between two batches of samples (Please see Figure R4)

Figure R4. (a) PXRD, (b) N_2 absorption and desorption isotherms (77 K), (c) pore size distribution for two batches of Zn-TAPB-COF-1.

Comment 3: Please explain why Zn-TAPT-COF-1 has the lowest specific surface area but the best catalytic performance in Table S3.

Response: As we can see from Figure 4b-f, Co-TAPT-COF-1 shows the best catalytic performance. In CO_2RR , the formation of $^*\text{COOH}$ intermediate from the hydrogenation of CO_2 is the rate-

determining step. Firstly, the excellent performance of Co-TAPT-COF-1 can be mainly attributed to the lowest reaction energy barrier of CO₂RR on Salen-Co site (0.75 eV) (Please see Table S10 in the revised Supporting Information). Secondly, the Co-TAPT-COF-1 with nanotube morphology and higher crystallinity possesses more efficient electron transfer in comparison with Co-TAPB-COF-1 and Co-TAPT-COF-2 (Please see Figure 4i in the revised Manuscript and Figure S48 in the revised Supporting Information). Thirdly, the nanotube morphology of Co-TAPT-COF-1 endows it higher CO₂ uptake capacity than Co-TAPT-COF-2 (Please see Figure S49 in the revised Supporting Information). All in all, the active sites, efficient electron transfer and high CO₂ uptake capacity jointly bring the best catalytic performance of Co-TAPT-COF-1. The influence of specific surface area on the catalytic performance is of secondary importance (Please see Table S5 in the revised Supporting Information).

Comment 4: Why does the synthesis route greatly affect the photocatalytic activity?

Response: The synthesis routes greatly affected the crystallinity and morphology of the M(salen)-COFs (Please see pages 6-11 in the revised Manuscript, Figure S7 and Figures S26-S33 in the revised Supporting Information). The M(salen)-COFs-1 obtained in one-step synthesis route showed higher crystallinities and larger-scale morphology than M(salen)-COFs-2 obtained from two-step synthesis. Steady-state photoluminescence spectra and nanosecond transient absorption spectra all indicated that M(salen)-COFs-1 with higher crystallinities exhibited longer promoted separation of electron-hole pairs, faster electron transfer and longer excited state lifetime. The above promoted properties greatly benefited photocatalytic activity (Please see pages 15-16 in the revised Manuscript). Moreover, the different morphologies of M(salen)-COFs from different synthetic routes also affected their CO₂ adsorption capacity and mass transfer efficiency (Please see Figure S49 in the revised Supporting Information). To make it more clear, we have rewritten the descriptions in the manuscript (Please see pages 15-16 in the revised Manuscript).

Comment 5: The model for calculating the free energy of CO₂ conversion to CO is too simple. It is just a metal ligand, and does not include the covalent structure in COF. It is recommended to redesign the calculation model to explain where the active center of the catalyst is.

Response: We thank the reviewer for pointing this out. We have redesigned the calculation models with more ligands (Figure R5 or Figure S42 in the revised Supporting Information). Free energy diagrams for CO₂ conversion to CO were calculated for different M(salen)-COFs (Please see Figure R6 and Figure S43 in Supporting Information). The calculated results indicated that Co-Salen site in Co-TAPT-COF showed the lowest rate-determining step (RDS) free energy. The RDS free energy on Zn-TAPT-COF and ZnZn-TAPT-COF were all higher than Co-TAPT-COF, confirming that the photocatalytic activities improved with lower RDS free energy (Please see Table S10 in the revised Supporting Information).

Figure R5. (Figure S42 in Supporting Information). Redesigned calculation models for (a) Zn-TAPT-COF, (b) ZnZn-TAPT-COF and (c) Co-TAPT-COF, respectively.

Figure R6 (Figure S43 in Supporting Information). Calculated free energy diagrams for CO₂ reduction to CO on Zn-TAPT-COF, ZnZn-TAPT-COF and Co-TAPT-COF.

Comment 6: The three catalysts in Fig. S3a have good absorption between 400-600 nm. Why do you add photosensitizer [Ru (bpy)₃] Cl₂ · 6H₂O.

Response: The [Ru(bpy)₃]Cl₂·6H₂O which can offer a large absorption coefficient for visible light functions as the photosensitizer in this work. In photocatalytic CO₂ reduction, the photosensitizer can harvest light and transfer the photoinduced electrons to the photocatalysts^[R3- R5]. As the M(salen)-COFs are heterogeneous catalysts and the light intensity decreases with distance from the solution surface, the light absorption by M(salen)-COFs catalysts are limited. So we add homogeneous photosensitizer [Ru(bpy)₃] Cl₂·6H₂O to enhance the visible light harvest.

Comment 7: It is suggested to supplement the circulating experimental data of the catalyst for CO₂ reduction to explain the chemical stability of the catalyst.

Response: We thank the editor for pointing this out. We have carried out the recycling experiments with Co-TAPT-COF-1 (stored for about one year), suggesting that there was no significant decrease in the catalytic performance over time (Please see Figure R7 or Figure S40 in Supporting Information). The stability of the Co-TAPT-COF-1 after photocatalysis was also assessed with PXRD patterns, FT-IR spectra and N₂ adsorption-desorption isotherms. The above characterizations proved that the crystallinity and covalent C=N linkage maintained after photocatalysis, while the BET surface area and pore volume decreased slightly (Please see Figure R8 or Figure S41 in Supporting Information).

Figure R7 (Figure S40 in Supporting Information). Stability test of Co-TAPT-COF-1 (stored for about one year) in five recycles for the photocatalytic CO₂ reduction reactions.

Figure R8 (Figure S41 in Supporting Information). Post-catalysis characterizations of the Co-TAPT-COF-1. (a) PXRD patterns, (b) FT-IR spectra, (c) N₂ adsorption-desorption isotherms (77 K) and (d) pore size distributions for Co-TAPT-COF-1 before and after photocatalysis.

Reviewer #2:

This manuscript describes the synthesis of COFs for CO₂ reduction. One big problem is that the PXRD patterns of COFs are inconsistent with their simulated or proposed structures. Another key issue is the experimentally observed pore size is far different from the proposed structure. The ‘one-pot’ method has been reported previously for a similar coordination system and is not novel. The catalytic activity and selectivity of the COFs are not impressive compared to previously reported other framework catalysts. Eye-catching phrases ‘Metallo-COFs’ and ‘single atom’ should be used carefully and avoided; ‘metallo-salen COFs’ is more accurate for this case. I cannot suggest the publication in Nat Commun.

Response: We thank the reviewer very much for the valuable comments. We have learned a lot from your points and this manuscript has been greatly improved based on your valuable comments and suggestions.

We have retreated the obtained M(salen)-COFs with solvent soaking and supercritical CO₂ (scCO₂) drying to remove the impurities in pores and maintain the mesopores of the M(salen)-COFs. After retreatment, the major peaks corresponding to (100) lattice plane of the M(salen)-COFs arose obviously and then the experimental PXRD patterns agreed well with their simulated AA stacking structures (Please see Figure R9 or Figure 2 in the revised Manuscript). Moreover, the pore sizes are also consistent with AA stacking structures after the retreatment (Please see Figures S5-S6 and Figures S18-S23 in the revised Supporting Information).

In this manuscript, M(salen)-COFs are used for photocatalytic CO₂ reduction to syngas for the first time, which will enrich the catalysts types for CO₂ photoreduction. The syngas production rate outperforms reported COFs. We are sorry that the title might mislead you and the “one-pot” is overemphasized. We have deleted the “One-step synthesis” in the title and changed the descriptions in the abstract to weaken this point (Please see page 1 in the revised Manuscript).

We have changed the phrases “metallo-COFs” into “metallo-salen-COFs (M(salen)-COFs)” (Please see the revised Manuscript and Supporting Information). We have changed the “single-atomic metal site and dual-atomic metal sites” to “mononuclear metal sites and binuclear metal sites”, respectively (Please see the revised Manuscript).

Figure R9. (a-c) 3D AA stacking structures, (d-f) TEM images and (g-i) experimental and simulated PXRD patterns for AA stacking and AB stacking modes (inset: simulated structure) of Zn-TAPB-COF-1, ZnZn-TAPB-COF-1 and Co-TAPT-COF-1, respectively.

Comment 1: In figure 1c, X should be CH or N.

Response: We thank the reviewer for pointing this out. We have changed the description in Figure 1 (Please see page 4 in the revised Manuscript).

Comment 2: The synthesis routes indicate that in figure 1 are quite confusing. Could you use a clearer layout or arrows to better convey the differences between these two methods?

Response: We thank the reviewer for this valuable suggestion. We have redrawn Figure 1 to better convey the differences between these two methods (Please see page 4 in the revised Manuscript).

Comment 3: In the legends of figure 2d-f, the black curve should be “Pawley-refined” instead of “simulated”.

Response: We thank the reviewer for pointing this out. We have rearranged the PXRD patterns to make it clear in Figure 2 (Please see Figure R9 or Figure 2 in page 4 of the revised Manuscript).

Comment 4: Could you show the exact peak position of the PXRD patterns or provide cif files of the simulated structures? The 2theta positions of the major peaks seem to be a bit too high if the COF structures are actually as drawn in figure 2a-c.

Response: As suggested by the reviewer, we have retreated the M(salen)-COFs samples with scCO_2 and retested the PXRD patterns of them. The exact peak positions of the PXRD patterns have been shown in Figure 2 and the major peaks with smaller 2theta positions corresponding to (100) lattice

planes of the COFs arose obviously. The cif files of the simulated structures have been also provided in the revised files.

Comment 5: In principle, the aldehyde groups can directly react with the tridentate amine ligands (TAPB/TAPT), which is a competition reaction of them reacting with ethylenediamine. If so, COF can also be produced. From the characterizations in this manuscript, it cannot rule out the possibility of these two reactions happening at the same time.

Response: We thank the reviewer for pointing this out. We have constructed the structure of COFs without ethylenediamine (TFB-TAPB-COF) [R6]. The simulated PXRD pattern of TFB-TAPB-COF is shown in Figure R10, which is totally different from the experimental PXRD of ZnZn-TAPB-1. While the experimental PXRD of ZnZn-TAPB-1 is consistent with the simulated AA stacking PXRD of ZnZn-TAPB-1, proving that ethylenediamine also react with the aldehyde groups. Moreover, the peaks around 65 ppm in ¹³C-NMR spectra of the M(salen)-COFs confirm the C-N from ethylenediamine, which act as necessary linker in the Salen-COFs (Please see Figures S13-S16) [R1-R2].

Figure R10. (a) Structure of TFB-TAPB-COF without ethylenediamine. (b) Experimental and simulated AA stacking PXRD of ZnZn-TAPB-1. Simulated PXRD pattern of TFB-TAPB-COF without ethylenediamine.

Comment 6: From the nitrogen sorption isotherms and pore size distributions, the pores of these M-COFs seem to be highly distributed, indicating the disordered structures.

Response: The highly distributed pores originate the residual impurities in the pores. We have retreated the COFs with solvent soaking and supercritical CO₂ (scCO₂) drying to remove the impurities in pores and maintain the mesopores of the M(salen)-COFs. The detailed descriptions have been added in the Methods section in the revised manuscript (Please see pages 17-19 in the revised Manuscript). After solvent soaking and scCO₂ drying, the pore size distributions of the

M(salen)-COFs-1 centred around 3.5 nm, agreeing well with the theoretical values in AA stacking structures (Please see Figures S5-S6 and Figures S18-S23 in the revised Supporting Information).

Comment 7: The influence of synthetic methods is frequently emphasized in this manuscript. However, all the contrasts between M-COF-1 to M-COF-2 can be attributed to their crystallinity difference (as shown in Figure S5). Amorphous counterparts should be put into comparisons.

Response: As suggested by the reviewer, we synthesized amorphous counterpart to Co-TAPT-COF-1 and used it as photocatalyst for CO₂ reduction. The photocatalytic activities, PXRD pattern and FT-IR spectrum of the amorphous counterpart were shown in Figure R11, indicating worse performance than the Co-TAPT-COF-1 and Co-TAPT-COF-2. This confirmed that the crystallinity of M(salen)-COFs significantly affected their performance.

Figure R11. Time-dependent production rate of (a) CO and (a) H₂ with 10 mg amorphous counterpart to Co-TAPT-COF-1 as catalyst. PXRD pattern and FT-IR spectrum of the amorphous counterpart to Co-TAPT-COF-1.

Comment 8: What is the stability of these COFs under photocatalytic conditions?

Response: We thank the reviewer for pointing this out. We have assessed the stability of the Co-TAPT-COF-1 after photocatalysis with PXRD patterns, FT-IR spectra and N₂ adsorption-desorption isotherms. The above characterizations proved that the crystallinity and covalent C=N linkage maintained after photocatalysis, while the BET surface area and pore volume decreased slightly (Please see Figure R8 or Figure S41 in Supporting Information).

Reviewer #3:

This manuscript described one-step synthesis of three M-COFs, which were further used as photocatalysts. Characterizations have been made to analyze the M-COFs and to evaluate their performance for CO₂ photoreduction. The topic is interesting, but the authors failed to demonstrate enough novelty and impact which can demand the publication in Nature Communications. The monomers and reactions are well-known. The crystallinities of the products do not reach the state of the art in this field. In addition to some flaws and claims without clear support, there are critical concerns about the modelled structures of the M-COFs, i.e., the layer stacking. Therefore, I cannot recommend its publication in Nature Communications. The detailed comments and questions are listed as following.

Response: We thank the reviewer very much for the valuable comments on our work. The crystallinities of the products have been significantly improved after solvent soaking and scCO₂ drying. The major peaks corresponding to (100) lattice plane of the COFs have arisen obviously (Please see Figure R9 or Figure 2 in the revised Manuscript). For the modelled structures, we have provided the simulated AA stacking and AB stacking PXRD patterns of the COFs (Please see Figure R9 or Figure 2 in the revised Manuscript). The peak positions and relative intensities of experimental line all coincided better with AA stacking than AB stacking. Moreover, the pore size detected by N₂ adsorption-desorption isotherms of the M(salen)-COFs were entirely consistent with AA stacking (Please see Figures S5-S6 and Figures S18-S23). We have made detailed modifications to the manuscript and responded to each question according to your comments. The specific contents are as follows:

Comment 1: In the abstract, the authors identified low crystallinities of COFs are the bottleneck of the field. While the statement is true, the high crystallinity of the COFs claimed by the authors in this study is misleading. The COFs only show less than ten peaks in the PXRD pattern, which is far from high crystallinity in the state of the art of M-COFs. See e.g., Nat. Commun. 2020, 11, 1434.

Response: We thank the reviewer for pointing this out. We have deleted “high crystallinity” in the revised Manuscript. Related reference (Nat. Commun. 2020, 11, 1434) has been cited in the revised Manuscript (Please see page 4 in the revised Manuscript).

Comment 2: The unit cell parameters and space groups in Figure 2 are inconsistent to those in Tables S1 and S2 and in the text.

Response: We thank the reviewer for pointing this out. We have modified the cell parameters and space groups in Figure 2 and Tables S2-S4 (Please see Figure 2 in the revised Manuscript and Tables S2-S4 in the revised Supporting Information).

Comment 3: The Pawley fitting shown in Figure 2 have serious issues. (i) There is no space group of P62m. (ii) For Figure 1d, there should be more Bragg positions before 5 degree for the space group P-62m and the given unit cell. (iii) Why the c-axis of ZnZn-TAPB-COF-1 is 19.99 Å while the structure model looks having the same layer stacking as the others? (iv) Why there is no peak at the first several Bragg positions?

Response: We thank the reviewer very much for pointing these issues out. The manuscript has been improved greatly based on your comments. (i) We have corrected the space group (Please see Figure 2 in the revised Manuscript and Tables S2-S4 in the revised Supporting Information). (ii) We have retreated the COFs with solvent soaking and supercritical CO₂ (scCO₂) drying to remove the impurities in pores and maintain the mesopores of the COFs. After retreatment, the major peaks

before 5 degree corresponding to (100) and (200) lattice planes of the COFs arose obviously (Please see Figure R9 or Figure 2 in the revised Manuscript). (iii) The unit cell parameters of ZnZn-TAPB-COF-1 have been modified (Please see Figure R9 or Figure 2 in the revised Manuscript and Table S3 in the revised Supporting Information). (iv) We have retreated the COFs with solvent soaking and scCO₂ drying to remove the impurities in pores and maintain the mesopores of the COFs. After retreatment, the major peaks before 5 degree corresponding to (100) and (200) lattice planes of the COFs arose obviously (Please see Figure R9 or Figure 2 in the revised Manuscript).

Comment 4: In modelling the structures of the COFs, the authors only considered the AA eclipsed stacking, which could cause misleading. How about other common layer stacking behaviors, such as AA inclined and AB staggered stacking?

Response: We thank the reviewer for this meaningful comment. We have added the simulated AA inclined and AB stacking structures for comparison (Please see Figure R12 and Figure 2 in the revised Manuscript). The peak positions and relative intensities of experimental line all coincided better with AA stacking than AA inclined or AB stacking. Moreover, the pore size of M-COF-1 detected by N₂ adsorption-desorption isotherms of the M(salen)-COFs were entirely consistent with AA stacking (Please see Figures S5-S6 and Figures S18-S23).

Figure R12. Simulated AA stacking, AA inclined, AB stacking PXRD patterns and experimental PXRD patterns for (a) Zn-TAPB-COF-1, (b) ZnZn-TAPB-COF-1, (c) Co-TAPT-COF-1.

Comment 5: To justify the structural models, I suggest to include a figure comparing experimental PXRD patterns with those calculated from the structural models.

Response: As suggested by the reviewer, we have added the simulated PXRD patterns in Figure 2 in the revised Manuscript.

Comment 6: How large are the pores measured from the structural models and how good do they agree with the N₂ sorption results? The pore size distributions in Figures S16-21 shows two pore sizes. One of c.a. 1.5 nm, and another of c.a. 3.0 nm. The authors need to describe where they come from.

Response: We have retreated the COFs with solvent soaking and supercritical CO₂ (scCO₂) drying to remove the impurities in pores and maintain the mesopores of the COFs. After retreatment, the pore sizes distributions were more centralized. The M(salen)-COFs-1 synthesized in one-step route showed high crystallinities and their pore sizes all centred around 3.5 nm, agreeing well with the theoretical pore sizes in AA stacking structures (Please see Figure 2 in the revised Manuscript, Figures S5-S6 and Figures S18-S23 in the revised Supporting Information). For M(salen)-COFs-2 synthesized in two-step route, their crystallinities were much weaker, indicating partial pore collapse. Hence the pore sizes of the M(salen)-COFs-2 centred between 1.1 nm and 1.4 nm due to partial pore collapse.

Comment 7: The authors mentioned the yield of ZnZn-TATP-COF-1 on line 130, and conclude the one step synthesis as having high yield. This is misleading. The authors need to show the yield of other one step made COFs and compare the yields to two step synthesis.

Response: As suggested by the reviewer, we have added yield of the COFs in the Methods section in the manuscript (Please see pages 17-19 in the revised Manuscript). The yields of the COFs are also listed for comparison (Please see Table R1 or Table S1 in the revised Supporting Information).

Table R1. Yields of M(salen)-COFs synthesized in different routes.

Samples	Zn-TAPB-COF		Zn-TAPT-COF		Co-TAPB-COF		ZnZn-TAPB-COF		ZnZn-TAPT-COF	
	One-step	Two-step	One-step	Two-step	One-step	Two-step	One-step	Two-step	One-step	Two-step
Yield (%)	87	67	67	66	83	53	81	39	75	36

Comment 8: On line 150, the authors claimed that one step synthesis show higher BET surface area than those synthesized by two steps. This is not true. Table S3 shows that half are the opposite.

Response: We thank the reviewer for pointing this out. We have deleted the description “the M(salen)-COFs synthesized by one-step method show higher S_{BET} than those synthesized by two-step methods in general” (Please see page 7 in the revised Manuscript).

Comment 9: The Zn-O bonding distance should be in the range of 1.9-2.1 Å for standard zinc oxides. The EXAFS results shown in Figure 3 show a distance of c.a. 1.6 Å, which has a significant deviation.

Response: The peak positions in FT-EXAFS spectra are normally 0.3-0.5 Å smaller than the true bonding distances, as the data are given without phase correction [R7-R9]. This deviation results from the phase shift ($\delta_j(K)$) in the formula below, which is used in the fourier transform. To make it clear, we have added “without phase correction” in the revised manuscript (Please see page 9 in the revised Manuscript).

$$\chi(K) = \sum_j \frac{N_j S_0^2 F_j(K) e^{-2R_j/\lambda(K)} e^{-2K^2 \sigma_j^2}}{K R_j^2} \sin[2K R_j + \delta_j(K)]$$

Comment 10: The section of morphological control of the COFs reads abruptly. Do the morphologies related to the focus of this manuscript, i.e., performance of CO₂ reduction?

Response: We thank the reviewer for this valuable comment. The morphologies of M(salen)-COFs are related to the electrons transfer behavior (Please see Figure 4i in the revised Manuscript and Figure S48 in the revised Supporting Information) and CO₂ absorption (Please see Figure S49 in the revised Supporting Information) in them, thus affect their photocatalytic activity (Figure 4e, f). To make the manuscript smoother, we have combined the section of morphological control with the section of synthesis and structural characterization of M(salen)-COFs to a section of synthesis and characterization of M(salen)-COFs. Figure 4 has been moved to SI (Please see pages 10-11 in the revised Manuscript and Figures S26-33 in the revised Supporting Information).

Comment 11: It is difficult to access the structural models from Tables S1 and S2. I suggest to upload cif files as supplementary information, even though they are modelled structures.

Response: As suggested by the reviewer, we have uploaded the cif files in the revised files.

Comment 12: The authors need to check the convention of writing space groups. For example, there is no space group should be written as P6/M.

Response: We thank the editor very much for pointing this out. We have checked the space groups and corrected the them in Figure 2 (Please see page 6 in the revised Manuscript and Tables S2-S4 in the revised Supporting Information).

Reviewer #4:

The synthesis of metallo-covalent organic frameworks (M-COFs) with high crystalline is highly desirable for their application in photocatalysis. This manuscript reported a very easy one-pot synthesis of highly crystalline M-COFs under ambient conditions and further demonstrated their application for syngas production from CO₂. The authors clearly characterized the structures of M-COFs with XRD, TEM, and EXAFS. Meanwhile, the authors demonstrated the generality of the methods for synthesizing other M-COFs. In photocatalysis, this M-COF disclosed a high syngas production rate. In overall, this work represents the new development in the synthesis and application of M-COFs. Considering the high novelty and significance, this Reviewer recommends the acceptance of this manuscript for publication in Nature Communications. There are some issues that should be addressed before publication.

Response: We thank Reviewer #4 for the positive comments.

Comment 1: Figure 1 presents the synthetic routes of M-COFs. However, it is not easy-to-understand. More text descriptions should be added in Figure 1.

Response: As suggested by the reviewer, we have redrawn Figure 1 to make it easier to understand (Please see page 4 in the revised Manuscript).

Comment 2: In this manuscript, alicylaldehyde was conducted as the ligand for the synthesis of M-COFs. Did the authors check other ligands or metal species? This would be helpful for the readers to synthesize other M-COFs.

Response: We thank the reviewer for this valuable comment. We have also synthesized M(salen)-COFs with nickel (Ni) salt via one-step synthesis. The synthesized Ni-TAPB-COF-1 also exhibited certain crystallinity (Please see Figure R1 in this letter).

Comment 3: As the metal species in the M-COFs are all in single-atomic or dual-atomic state, the TOF of CO production should be given and compared with references.

Response: We thank the editor for this valuable suggestion. We have added the TOF of CO production in the revised manuscript as “The calculated turnover frequencies of CO and H₂ are 4.88 h⁻¹ and 6.22 h⁻¹ in 4 h, respectively.” (Please see page 13 in the revised Manuscript).

Comment 4: How about the structural stability of the M-COFs? This would be very important for their application in catalysis.

Response: We thank the reviewer for pointing this out. The stability of the Co-TAPT-COF-1 after photocatalysis was assessed with PXRD patterns, FT-IR spectra and N₂ adsorption-desorption isotherms. The above characterizations proved that the crystallinity and covalent C=N linkage maintained after photocatalysis, while the BET surface area and pore volume decreased slightly (Please see Figure R8 or Figure S41 in the revised Supporting Information).

References

- [R1] Li, L. H. et al. Salen-based covalent organic framework. *J. Am. Chem. Soc.* **139**, 6042-6045 (2017).
- [R2] Huang, J. J. et al. Microporous 3D covalent organic frameworks for liquid chromatographic separation of xylene isomers and ethylbenzene. *J. Am. Chem. Soc.* **141**, 8996-9003 (2019).
- [R3] Zhang, Q. et al. Designing covalent organic frameworks with Co-O₄ atomic sites for efficient CO₂ photoreduction. *Nat. Commun.* **14**, 1147 (2023).
- [R4] Gao, C. et al. Heterogeneous single-atom catalyst for visible-light driven high-turnover CO₂ reduction: the role of electron transfer. *Adv. Mater.* **30**, 1704624 (2018).
- [R5] Ran, L. et al. Engineering Single-Atom Active Sites on Covalent Organic Frameworks for Boosting CO₂ Photoreduction. *J. Am. Chem. Soc.* **144**, 17097-17109 (2022).
- [R6] Xiong, Z. S. et al. Amorphous-to-crystalline transformation: General synthesis of hollow structured covalent organic frameworks with high crystallinity. *J. Am. Chem. Soc.* **144**, 6583-6593 (2022).
- [R7] Jia, Y. L. et al. Tailoring the Electronic Structure of an Atomically Dispersed Zinc Electrocatalyst: Coordination Environment Regulation for High Selectivity Oxygen Reduction. *Angew. Chem. Int. Ed.* **61**, e20211083 (2022).
- [R8] Yang, Q. H. et al. Metal–Organic-Framework-Derived Hollow N-Doped Porous Carbon with Ultrahigh Concentrations of Single Zn Atoms for Efficient Carbon Dioxide Conversion. *Angew. Chem. Int. Ed.* **58**, 3511-3515 (2019).
- [R9] Yang, H. Z. et al. A universal ligand mediated method for large scale synthesis of transition metal single atom catalysts. *Nat. Commun.* **10**, 4585 (2019).

REVIEWER COMMENTS

Reviewer #1 (Remarks to the Author):

I have carefully reviewed the revised version of the manuscript and Point-by-point Response to the Comments. Basically met the revision requirements and agreed to be published.

Reviewer #2 (Remarks to the Author):

The author did revise certain minor issues in this manuscript. However, some major problems are still not addressed. The novelty of this method and the catalytic performance are still not impressive enough. Therefore, I cannot suggest publication in Nat Commun.

Reviewer #3 (Remarks to the Author):

I appreciate the authors' efforts on revising the manuscript, which has been improved. However, concerns still remain on the proposed structural models of the COFs. While the AA eclipsed model can be distinguished from AB staggered model by PXRD and N₂ sorption results, the authors failed to describe why the AA eclipsed model can be distinguished from AA inclined model and AA serrated model. The detailed comments and questions are listed as following.

1. The authors only presented the discussion of AA eclipsed model and AB staggered model in the manuscript. As different stacking behaviors can greatly affect the properties of 2D-COFs (see ACS Appl. Nano Mater. 2022, 5, 10, 14377), the authors need to describe how the AA eclipsed model can be distinguished from other stacking models, i.e., AA inclined and AA serrated models in the main text.
2. The authors need to describe in the manuscript how it was found that "The peak positions and relative intensities of experimental line all coincided better with AA stacking than AA inclined". Due to the low number of peaks in the experimental pattern, it is difficult to distinguish them.
3. Only 1D fringes were observed by TEM imaging in Figure 2. This is not convincing to support the proposed eclipsed AA-stacking models, which have honeycomb structures. The 1D fringes actually agree better with AA inclined models or AA serrated models, which do not have 6-fold or 3-fold symmetry. I suggest the authors to investigate the details of the TEM images, as well as their FFT, which can provide important structural information. In addition, the author need to mention from which direction the images are recorded.
4. As 001 reflection is not obvious from PXRD pattern, how did the authors determine the inter-layer distance (c-axis) of the models? Why does Zn-TAPB-COF-1 have a much longer c-axis than the others?
5. The authors need to report corrected EXAFS results because they can give indication not only about Zn-O distance, but also Zn-Zn distance, which is related to the inter-layer distance of the COFs (see my comment above).

Reviewer #4 (Remarks to the Author):

In this version, the authors have properly responded the comments from the reviewers and revised the manuscript well. This reviewer suggests the acceptance of this manuscript for publication now.

Reviewer #5 (Remarks to the Author):

The design, synthesis and structural characterization of this work are interesting. The X-ray absorption

spectroscopy results are convincing. However, the photophysical experiments and the subsequent discussions raise concerns. Using Ru(bpy)₃ in photocatalytic experiments amplifies these issues further.

1) Firstly, no experimental details of photoluminescence and nanosecond transient absorption measurements were provided.

2) Furthermore, the results derived from the steady-state photoluminescence and the nanosecond absorption spectroscopy do not corroborate the proposed photocatalytic performance. First of all, the authors did not report what the photoluminescence is originated from. Is it due to COFs or the metallic segment of the structure? Which unit is the emission quencher?

3) There is significant issue concerning the nanosecond transient absorption spectra. The authors only displayed the kinetics at a wavelength of 500 nm. Without clarification about what this 500 nm signal denotes, the use of the lifetime to substantiate the charge separation efficiency can be misleading. This issue is further complicated when Ru(bpy)₃ is employed as a photosensitizer. This is because, in this context, both the lifetime of Ru(bpy)₃ and the charge transfer process from Ru(bpy)₃ to M-COF become crucial factors influencing the photocatalytic efficiency. In another word, the excited state dynamics from COF may not be important.

Point-by-point Response to the Reviewers' Comments

Reviewer #1:

Comment: I have carefully reviewed the revised version of the manuscript and Point-by-point Response to the Comments. Basically met the revision requirements and agreed to be published.

Response: We thank Reviewer #1 for the positive comments.

Reviewer #2:

Comment: The author did revise certain minor issues in this manuscript. However, some major problems are still not addressed. The novelty of this method and the catalytic performance are still not impressive enough. Therefore, I cannot suggest publication in Nat Commun.

Response: We would like to thank Reviewer #2 for the valuable comments. In the following, we will elaborate on the major issues we have addressed in the revisions, discuss the novelty of this method, as well as the catalytic performance. We hope this addresses your concerns.

A. All raised questions have helped us in improving the quality of the manuscript. Major issues that have been addressed during the first round of revisions are as follows.

- a) We reported the quantum yields, turnover frequencies and numbers, recycling experiments and stability assessments of the catalysts with post-catalysis characterization of their structures.
- b) We proved that COFs with ethylenediamine were generated instead of MOFs through PXRD and ^{13}C -NMR spectra.
- c) We redesigned the DFT calculation models and re-calculated the free energy diagrams.
- d) We treated the M(salen)-COFs with supercritical CO_2 (scCO_2) drying. Subsequently, the experimental PXRD patterns, pore sizes all agreed well with their simulated structures. Additionally, the consistency of BET data of M(salen)-COFs improved.

e) We corrected the errors in structural refinement, and cif files were uploaded as
supplementary information.

f) We improved the experimental methods sections.

**B. The novelty of this method are as follows.**

a) This synthetic strategy for M(salen)-COFs proposed in this work was general,
facile, efficient, and scalable, yielding quantities ranging from 2.61g to 3.56g.
Conducting the synthesis in autoclaves, without the need for evacuation and
deoxygenation, significantly simplifies the production of M(salen)-COFs. For the
M(salen)-COFs synthesis, in most cases, the ampoule containing ligands, solvents
and catalysts need to be flash frozen in a liquid nitrogen bath, evacuated and flame
sealed^[R1-R2]. In many situations, two-step processes involving post-synthetic
metalation are necessary^[R3]. These methods typically yield only tens of milligrams
M(salen)-COFs^[R1-R3]. Thus, large-scale and facile preparation of M(salen)-COFs
remains a significant challenge.

b) In this work, we are the first to report on crystalline M(salen)-COFs with dual-
atomic sites. Single-atomic metal site and dual-atomic sites in M(salen)-COFs
could be easily achieved by adjusting the coordination environments. This
represented a significant advancement, elevating the facile controllability of
M(salen)-COFs to a new level.

c) We introduced supercritical CO₂ treatment to tackle the issue of crystalline
structure collapse during the synthesis of mesoporous M(salen)-COFs.

51 d) Co-TAPT-COF-1, synthesized through the one-step reaction, self-assembled into
52 hollow nanotubes featuring a macropore channel (inner diameter of 135 nm and
53 outer diameter of 220 nm). This was the first example for the self-assembled
M(salen)-COF nanotubes.

**C. Catalytic performance.**

a) In this work, M(salen)-COFs were introduced for the first time as heterogenous
catalysts in CO₂ photoreduction. M(salen) is crucially important in coordination
chemistry, being recognized as one of the most powerful homogeneous catalysts.
By embedding M(salen) within M(salen)-COFs, we can harness the unique

catalytic activities of M(salen) while facilitating easier catalyst recovery. While
M(salen)-COFs possess exceptional photosensitivity, making them highly
promising for CO₂ photoreduction, no report on this application have been
presented to date. Our work demonstrated that M(salen)-COFs exhibited superior
photocatalytic activities in CO₂ reduction. We anticipate an increase in studies
exploring the applications of M(salen)-COFs in photocatalytic CO₂ conversion in
the future.

b) This work comprehensively explored the influence of synthesis routes, metal
species and coordination environment, ligands of M(salen)-COFs on their
photocatalytic CO₂ activities. M(salen)-COFs derived from one-pot synthesis
demonstrated enhanced photocatalytic performance, attributed to their higher
crystallinity and superior carrier behavior. We believe that our findings will pave
the way for the development of high-activity catalysts for photocatalytic CO₂
reduction in upcoming research.

**Reviewer #3:**

**Comment:** I appreciate the authors' efforts on revising the manuscript, which has been
improved. However, concerns still remain on the proposed structural models of the
COFs. While the AA eclipsed model can be distinguished from AB staggered model by
PXRD and N₂ sorption results, the authors failed to describe why the AA eclipsed model
can be distinguished from AA inclined model and AA serrated model. The detailed
comments and questions are listed as following.

**Response:** We are grateful to Reviewer #3 for the positive feedback and constructive
suggestions. All raised questions have greatly improved the quality of the manuscript.
In this revision, we have re-optimized the structural models of the COFs. Both AA
inclined model and AA serrated model have been added for detailed comparative
analysis (Please see Figure 2 in the revised Manuscript). Notably, the experimental
PXRD patterns of all three M(salen)-COFs aligned more closely with their eclipsed
AA-stacking and serrated AA-stacking models. Given that no honeycomb structures
were observed in their TEM images, AA serrated models without 6-fold or 3-fold

symmetry were more plausible modes for the synthesized M(salen)-COFs. We have
 rewritten the descriptions in the manuscript (Please see pages 6-7 in the revised
 Manuscript).

 **Figure 2.** (a-c) Serrated AA-stacking structures, (d-f) TEM images and (g-i)
 experimental and simulated PXRD patterns of Zn-TAPB-COF-1, ZnZn-TAPB-COF-1
 and Co-TAPT-COF-1, respectively.

**Comment 1:** The authors only presented the discussion of AA eclipsed model and AB
 staggered model in the manuscript. As different stacking behaviors can greatly affect
 the properties of 2D-COFs (see ACS Appl. Nano Mater. 2022, 5, 10, 14377), the authors
 need to describe how the AA eclipsed model can be distinguished from other stacking
 models, i.e., AA inclined and AA serrated models in the main text.

**Response:** We thank the reviewer for this valuable suggestion. We have carefully
 analyzed the simulated PXRD patterns of eclipsed AA-stacking, serrated AA-stacking,
 inclined AA-stacking and AB-stacking models for M(salen)-COFs (Please see Figure 2
 in the revised Manuscript and Figures S5-S6 in the revised Supporting Information).
 Detailed descriptions have been improved as “Simulated eclipsed AA-stacking, serrated
 AA-stacking, inclined AA-stacking and AB-stacking models for Zn-TAPB-COF-1,

ZnZn-TAPB-COF-1 and Co-TAPT-COF-1 were constructed using Materials Studio
software (Figures 2g-i and Figures S5-S6). The experimental PXRD patterns of the
three M(salen)-COFs were all in better consistency with their eclipsed AA-stacking and
serrated AA-stacking models. Given that only 1D fringes were observed and
honeycomb structures were absent in their TEM images, AA serrated models without
6-fold or 3-fold symmetry were more reasonable modes for the synthesized M(salen)-
COFs.” (Please see page 6 in the revised Manuscript). We have cited this reference
(ACS Appl. Nano Mater. 2022, 5, 10, 14377) in the revised Manuscript.

**Comment 2:** The authors need to describe in the manuscript how it was found that “The
peak positions and relative intensities of experimental line all coincided better with AA
stacking than AA inclined”. Due to the low number of peaks in the experimental pattern,
it is difficult to distinguish them.

**Response:** We thank the reviewer for pointing this out. In response, we have added the
simulated PXRD patterns of inclined AA-stacking for a more detailed comparison
(Please see Figure 2 in the revised Manuscript). To enhance clarity, auxiliary lines have
been added in the PXRD patterns, highlighting the correspondence of peak positions.
Upon examination, it’s evident that the experimental peak positions of M(salen)-COFs
align more closely with the eclipsed AA-stacking or serrated AA-stacking than with the
inclined AA-stacking.

**Comment 3:** Only 1D fringes were observed by TEM imaging in Figure 2. This is not
convincing to support the proposed eclipsed AA-stacking models, which have
honeycomb structures. The 1D fringes actually agree better with AA inclined models
or AA serrated models, which do not have 6-fold or 3-fold symmetry. I suggest the
authors to investigate the details of the TEM images, as well as their FFT, which can
provide important structural information. In addition, the author need to mention from
which direction the images are recorded.

**Response:** We thank the reviewer for this valuable comment. As suggested by the
reviewer, we investigated the details of TEM images on three separate occasions using
different devices. Despite our thorough investigations, we were unable to identify any
honeycomb pore structures. Consequently, the AA serrated models without 6-fold or 3-

137 fold symmetry emerge as the most plausible modes for the synthesized M(salen)-COFs.
We have rewritten the descriptions in the manuscript as “Simulated eclipsed AA-
stacking, serrated AA-stacking, inclined AA-stacking and AB-stacking models for Zn-
TAPB-COF-1, ZnZn-TAPB-COF-1 and Co-TAPT-COF-1 were constructed using
Materials Studio software (Figures 2g-i and Figures S5-S6). The experimental PXRD
patterns of the three M(salen)-COFs were all in better consistency with their eclipsed
AA-stacking and serrated AA-stacking models. Given that only 1D fringes were
observed and honeycomb structures were absent in their TEM images, AA serrated
models without 6-fold or 3-fold symmetry were more reasonable modes for the
synthesized M(salen)-COFs.” (Please see page 6 in the revised Manuscript). Diffraction
spots in FFT were also not captured due to the destruction of the crystal form by high-
voltage electron beam.

**Comment 4:** As 001 reflection is not obvious from PXRD pattern, how did the authors
determine the inter-layer distance (c-axis) of the models? Why does Zn-TAPB-COF-1
have a much longer c-axis than the others?

**Response:** The inter-layer distances (c-axis) were determined through optimization
using a universal force field, rather than from 001 reflections in experimental PXRD
patterns. Upon reviewing the structure, we noted that the Salen moiety in Zn-TAPB-
COF-1 was twisted in the previous version. After re-optimizing the structure of the
Salen(M)-COFs, the c-axis of Zn-TAPB-COF-1 now aligns more closely with the other
structures (Please see pages 6-7 in the revised Manuscript.)

**Comment 5:** The authors need to report corrected EXAFS results because they can
give indication not only about Zn-O distance, but also Zn-Zn distance, which is related
to the inter-layer distance of the COFs (see my comment above).

**Response:** As suggested by the reviewer, we have updated the EXAFS results (Please
see Figure 3 in the revised Manuscript) in the main text as “The FT-EXAFS spectrum
of Zn foil showed major peak at 2.64 Å (with phase correction), which was assigned to
Zn-Zn bonds. Meanwhile, ZnO exhibited two main peaks at 1.95 Å and 3.25 Å
assigning to Zn-O and Zn-Zn bonds, respectively. FT-EXAFS spectra of both Zn-
TAPB-COF-1 and ZnZn-TAPB-COF-1 showed main peaks around 1.94 Å,

corresponding to Zn-O or Zn-N bond lengths (Figures 3d and 3e)^{30,32}. No signature of
 Zn-Zn bonds was observed in the above Zn-COFs, validating the absence of Zn
 nanoparticles or clusters. Furthermore, the EXAFS fitting results confirmed that the Zn
 atom was coordinated with approximately two N atoms and two O atoms in both Zn-
 TAPB-COF-1 and ZnZn-TAPB-COF-1 (Table S6). The FT-EXAFS spectra of Co foil,
 CoO and Co₃O₄ showed major peaks around 2.48 Å, 2.99 Å and 2.86 Å (Figure 3f),
 corresponding to Co-Co and Co-O bonds. As for Co-TAPT-COF-1, two main peaks at
 1.86 Å and 2.61 Å were assigned to Co-O (N) bonds from salen moiety and Co-O bonds
 from H₂O.” (Please see page 10 in the revised Manuscript). No obvious signatures of
 Zn-Zn bonds were observed in the above Zn-COFs.

 **Figure 3.** AC-TEM images for (a) Zn-TAPB-COF-1, (b) ZnZn-TAPB-COF-1 and (c)
 Co-TAPT-COF-1 respectively. (d) FT-EXAFS spectra for Zn R-space of ZnO, Zn foil
 and Zn-TAPB-COF-1. (e) FT-EXAFS spectra for Zn R-space of ZnO, Zn foil and ZnZn-
 TAPB-COF-1. (f) FT-EXAFS spectra for Co R-space of CoO, Co₃O₄, Co foil and Co-
 TAPT-COF-1. (h-i) Wavelet transform (WT) for Zn-TAPB-COF-1, ZnZn-TAPB-COF-
 1 and Co-TAPT-COF-1, respectively.

**Reviewer #4:**

**Comment:** In this version, the authors have properly responded the comments from the
reviewers and revised the manuscript well. This reviewer suggests the acceptance of
this manuscript for publication now.

**Response:** We thank Reviewer #4 for the positive comments.

**Reviewer #5:**

**Comment:** The design, synthesis and structural characterization of this work are
interesting. The X-ray absorption spectroscopy results are convincing. However, the
photophysical experiments and the subsequent discussions raise concerns. Using
Ru(bpy)₃ in photocatalytic experiments amplifies these issues further.

**Response:** We thank Reviewer #5 for the positive comments and suggestions. We have
modified the photophysical experiments and subsequent discussions (Please see page
16 in the revised Manuscript). We understand your concerns about using Ru(bpy)₃ in
photocatalytic experiments. As the M(salen)-COFs are heterogeneous catalysts and the
light intensity decreases with distance from the solution surface, the light absorption by
M(salen)-COFs heterogeneous catalysts is limited. So we add homogeneous
photosensitizer [Ru(bpy)₃] Cl₂·6H₂O to enhance the visible light harvest^[R4].

**Comment 1:** Firstly, no experimental details of photoluminescence and nanosecond
transient absorption measurements were provided.

**Response:** As suggested by the reviewer, we have added experimental details of
photoluminescence in the part of Methods as “Steady-state photoluminescence spectra
was conducted using an EDINBURGH FLS1000 Steady-state/Lifetime
Spectrofluorometer, with an excitation wavelength of 280 nm. The emission
wavelength range spanned from 300 to 900 nm. The integration time was 0.2 seconds,
and the excitation wavelength slit width was set at 4 nm. The emission wavelength slit
width was set at 4 nm as well. The step size was 1 nm.” (Please see page 20 in the
revised Manuscript). The nanosecond transient absorption measurements have been
removed from this work due to the low signal-to-noise ratio and complex reaction
system (Please see the response to Comment 3).

**Comment 2:** Furthermore, the results derived from the steady-state photoluminescence
and the nanosecond absorption spectroscopy do not corroborate the proposed
photocatalytic performance. First of all, the authors did not report what the
photoluminescence is originated from. Is it due to COFs or the metallic segment of the
structure? Which unit is the emission quencher?

**Response:** We thank the reviewer for pointing this out. We have conducted steady-state
photoluminescence spectra of the organic ligands and metallic segment separately to
penetrate the origination of photoluminescence (Please see Figure S48 in the revised
Supporting Information). As we can see, photoluminescence originated from TAPB or
TAPT ligand, while the metallic segment (Co-Salen) was the emission quencher^[R5].
Significantly, Co-TAPT-COF-1 showed no fluorescence. This complete fluorescence
quenching in Co-TAPT-COF-1 demonstrated promoted electron-hole separation due to
its higher crystallinity and nanotube morphology. Co-TAPT-COF-2 and Co-TAPB-
COF-1 both showed certain fluorescence originating from electron-hole recombination.
The steady-state photoluminescence results corroborated their photocatalytic
performance well (Please see Figure 4 in the revised Manuscript). The catalyst with
stronger fluorescence showed lower photocatalytic activity.

**Figure S48.** Steady-state photoluminescence spectra of TAPT ligand, TAPB ligand, and
Co-Salen.

**Comment 3:** There is significant issue concerning the nanosecond transient absorption
spectra. The authors only displayed the kinetics at a wavelength of 500 nm. Without
clarification about what this 500 nm signal denotes, the use of the lifetime to

substantiate the charge separation efficiency can be misleading. This issue is further
complicated when Ru(bpy)₃ is employed as a photosensitizer. This is because, in this
context, both the lifetime of Ru(bpy)₃ and the charge transfer process from Ru(bpy)₃ to
M-COF become crucial factors influencing the photocatalytic efficiency. In another
word, the excited state dynamics from COF may not be important.

**Response:** We thank the reviewer for this valuable comment. We retested the transient
absorption spectroscopy, but the signal-to-noise ratio was still poor. Considering that
the excited state dynamics from COF might not be significant due to the presence of
Ru(bpy)₃, we have decided to remove the transient absorption data and associated
analysis (Please see page 16 in the revised Manuscript and Figure S48 in the revised
Supporting Information).

**References**

[R1] Huang, J. J. et al. Microporous 3D covalent organic frameworks for liquid
chromatographic separation of xylene isomers and ethylbenzene. *J. Am. Chem. Soc.*
**141**, 8996 (2019).

[R2] Han, X. et al. Chiral covalent organic frameworks with high chemical stability for
heterogeneous asymmetric catalysis. *J. Am. Chem. Soc.* **139**, 8693 (2017).

[R3] Li, L. H. et al. Salen-based covalent organic framework. *J. Am. Chem. Soc.* **139**,
6042 (2017).

[R4] Zhang, Q. et al. Designing covalent organic frameworks with Co-O₄ atomic sites
for efficient CO₂ photoreduction. *Nat. Commun.* **14**, 1147 (2023).

[R5] DeVore, M. A. et al. Characterization of Quinoxolinol Salen Ligands as Selective
Ligands for Chemosensors for Uranium. *Eur. J. Org. Chem.* **34**, 5708 (2016).

REVIEWERS' COMMENTS

Reviewer #3 (Remarks to the Author):

The authors have satisfactorily addressed my comments. I recommend the manuscript for publication.

Reviewer #5 (Remarks to the Author):

I appreciate the authors' efforts in revising the manuscript according to my comments earlier. It is good idea to remove nanosecond transient absorption results because the results did not reflect the charge separation dynamics between Ru(bpy)₃ and COFs. Given the utilization of Ru(bpy)₃ as the photosensitizer, the nanosecond absorption findings solely pertain to the excited state/carrier dynamics of COFs, which may not necessarily correlate with their catalytic performance. Similarly, I recommend the removal of the steady-state emission results, as they primarily pertain to the excited state/charge transfer/recombination within COFs, a factor potentially less significant when Ru(bpy)₃ serves as the photosensitizer. It is evident that the authors lack a comprehensive understanding of the catalytic mechanism within the Ru(bpy)₃/COFs photocatalytic system, which differs significantly from the system lacking Ru(bpy)₃. Incorporating results that do not align with their conclusion has a detrimental effect on the credibility of their work.

Point-by-point Response to the Reviewers' Comments

Reviewer #3:

Comment: The authors have satisfactorily addressed my comments. I recommend the manuscript for publication.

Response: We thank Reviewer #3 for the positive comments.

Reviewer #5:

Comment: I appreciate the authors' efforts in revising the manuscript according to my comments earlier. It is good idea to remove nanosecond transient absorption results because the results did not reflect the charge separation dynamics between Ru(bpy)₃ and COFs. Given the utilization of Ru(bpy)₃ as the photosensitizer, the nanosecond absorption findings solely pertain to the excited state/carrier dynamics of COFs, which may not necessarily correlate with their catalytic performance. Similarly, I recommend the removal of the steady-state emission results, as they primarily pertain to the excited state/charge transfer/recombination within COFs, a factor potentially less significant when Ru(bpy)₃ serves as the photosensitizer. It is evident that the authors lack a comprehensive understanding of the catalytic mechanism within the Ru(bpy)₃/COFs photocatalytic system, which differs significantly from the system lacking Ru(bpy)₃. Incorporating results that do not align with their conclusion has a detrimental effect on the credibility of their work.

Response: We thank Reviewer #1 for the positive comments. As suggested by the reviewer, we have removed the steady-state emission results (Please see Figure 4i and page 16 in the revised Manuscript and Figure S48 in the revised Supporting Information).